# DualToken: Towards Unifying Visual Understanding and Generation with Dual Visual Vocabularies

**Wei Song**[1,2,3] **Yuran Wang**[5] **Zijia Song**[3] **Yadong Li**[4]
**Zenan Zhou**[4] **Long Chen**[6] **Jianhua Xu**[4*†] **Jiaqi Wang**[2*] **Kaicheng Yu**[1,2,3*]

[1]Zhejiang University  [2]Shanghai Innovation Institute  [3]Westlake University
[4]Baichuan Inc.  [5]Peking University  [6]Xiaomi EV

jhxu_org@163.com, wjqdev@gmail.com
{songwei, kyu}@westlake.edu.cn

## Abstract

The differing representation spaces required for visual understanding and generation pose a challenge in unifying them within the autoregressive paradigm of large language models. A vision tokenizer trained for reconstruction excels at capturing low-level visual appearance, making it well-suited for visual generation but lacking high-level semantic representations for understanding tasks. Conversely, a vision encoder trained via contrastive learning aligns well with language but struggles to decode back into the pixel space for generation tasks. To bridge this gap, we propose **DualToken**, a method that unifies representations for both understanding and generation within a single tokenizer. However, directly integrating reconstruction and semantic objectives creates conflicts, leading to degraded performance in both reconstruction fidelity and semantic accuracy. Instead of forcing a single codebook to capture both visual appearance and semantics, DualToken disentangles them by introducing separate codebooks for high-level semantics and low-level visual details. As a result, DualToken achieves 0.25 rFID and 82.0% zero-shot accuracy on ImageNet, and demonstrates strong effectiveness in downstream MLLM tasks for both understanding and generation. Specifically, our method surpasses VILA-U by 5.8 points on average across ten visual understanding benchmarks and delivers a 13% improvement on GenAI-Bench. Notably, incorporating dual visual tokens outperforms using a single token type on both understanding and generation tasks. We hope our research offers a new perspective on leveraging dual visual vocabularies for building unified vision–language models. Project page is available here.

## 1 Introduction

Unifying visual understanding and generation within the pure autoregressive (AR) paradigm of Large Language Models (LLMs) offers a simple, end-to-end alternative to the increasingly common yet structurally complex approach of coupling LLMs with external diffusion modules (Dong et al., 2024; Huang et al., 2025; Pan et al., 2025; Chen et al., 2025b). To enable fully unified AR modeling of vision and language, a model requires a visual tokenizer to map images into discrete tokens and a corresponding detokenizer that can faithfully reconstruct them back into pixel space.

Early methods in this direction (Yu et al., 2023a; Team, 2024; Wang et al., 2024b) directly adopt the encoder and decoder of VQ-VAE as the visual tokenizer and detokenizer. While these approaches demonstrated the feasibility of unifying visual understanding and generation within the AR paradigm, their understanding capabilities are typically lacking compared to multimodal large language models (MLLMs) specialized for understanding tasks (Liu et al., 2023; Yue et al., 2023; Fu et al., 2024; Song et al., 2024). We argue that this performance gap stems from inadequate visual representations: traditional VQ-VAEs are optimized solely for reconstruction, producing image tokens that preserve

---

*Corresponding authors.
†Project Leader.

Figure 1: **(Left)** Challenges faced by existing visual tokenizers. **(Middle)** We compare zero-shot classification accuracy and reconstruction FID on ImageNet-1K(val) across baseline methods and DualToken. DualToken achieves results comparable to or surpassing both semantic-only and reconstruction-only methods in both tasks. **(Right)** Reconstruction results of VILA-U and DualToken, our DualToken significantly outperforms VILA-U, which suffers from severe distortion and blurriness.

low-level visual details but fail to capture high-level semantics aligned with language. By contrast, MLLMs designed for understanding tasks (Liu et al., 2024c; Chen et al., 2024b; Li et al., 2024c; 2025b; Bai et al., 2025) typically rely on CLIP-family encoders (Radford et al., 2021; Zhai et al., 2023), which are pretrained with text alignment and thus inherently encode high-level semantics, making them more suitable for downstream visual understanding tasks in MLLMs.

To fully leverage the language-aligned semantic representations of CLIP, a natural approach is to quantize the features of a CLIP encoder and train a decoder for image reconstruction (Wu et al., 2025). This involves learning to reconstruct images for downstream generation tasks while preserving its semantic capabilities as much as possible (Wu et al., 2025). However, as shown in Fig.1 and Table.1, directly combining reconstruction and semantic objectives often leads to severe distortions and blurriness in reconstruction tasks, along with a noticeable decline in semantic metrics such as zero-shot classification and image-text retrieval, compared to its original pretrained model (Zhai et al., 2023). This degradation, as discussed in Wu et al. (2025), reflects the inherently conflict between the two training objectives, ultimately limiting both the quality of downstream image generation tasks and the performance of multimodal understanding tasks.

Table 1: **Comparison to state-of-the-art visual tokenizers.** DualToken achieves the best performance among existing unified visual tokenizers in semantic metrics. It also mitigates the distortion and blurriness faced by VILA-U during reconstruction, and surpasses dedicated models in reconstruction metrics.

| METHODS | Semantic | | | Reconstruction | | |
|---|---|---|---|---|---|---|
| | Zero-Shot[↑] | T2I(R@1)[↑] | I2T(R@1)[↑] | rFID[↓] | PSNR[↑] | SSIM[↑] |
| *Reconstruction Only* | | | | | | |
| MoVQGAN (Zheng et al., 2022) | ✗ | ✗ | ✗ | 1.12 | 22.42 | 0.673 |
| RQ-VAE (Lee et al., 2022) | ✗ | ✗ | ✗ | 2.69 | - | - |
| ViT-VQGAN (Yu et al., 2021) | ✗ | ✗ | ✗ | 1.55 | - | - |
| Open-MAGVIT2 (Luo et al., 2024) | ✗ | ✗ | ✗ | 1.17 | 21.90 | - |
| SBER-MoVQGAN (SberBank, 2023) | ✗ | ✗ | ✗ | 0.68 | 27.04 | 0.741 |
| *Understanding Only* | | | | | | |
| CLIP-L/14-336 (Radford et al., 2021) | 76.6 | 21.2 | 21.5 | ✗ | ✗ | ✗ |
| SigLIP-L/16-256 (Zhai et al., 2023) | 80.5 | 21.0 | 21.4 | ✗ | ✗ | ✗ |
| SigLIP-So/14-384 (Zhai et al., 2023) | 83.2 | 21.7 | 21.6 | ✗ | ✗ | ✗ |
| SigLIP2-So/16-256 (Tschannen et al., 2025) | 83.4 | 21.5 | 22.0 | ✗ | ✗ | ✗ |
| ViTamin-L/16-256 (Chen et al., 2024a) | 81.2 | 20.6 | 21.2 | ✗ | ✗ | ✗ |
| *Reconstruction & Understanding* | | | | | | |
| QLIP (256px) (Zhao et al., 2025) | 74.3 | 16.8 | 18.4 | 3.21 | 23.16 | 0.628 |
| QLIP (392px) (Zhao et al., 2025) | 79.1 | 20.4 | 21.0 | 1.46 | 25.36 | 0.690 |
| UniTok (Ma et al., 2025) | 78.6 | - | - | 0.38 | 25.34 | - |
| TokenFlow (256px) (Qu et al., 2024) | - | - | - | 1.37 | 21.41 | 0.687 |
| TokenFlow (384px) (Qu et al., 2024) | - | - | - | 0.63 | 22.77 | 0.731 |
| Muse-VL (256px) (Xie et al., 2025b) | - | - | - | 2.26 | 20.14 | 0.646 |
| TokLIP (SigLIP-So/16-256) (Lin et al., 2025) | 80.0 | - | - | 0.94 | 21.94 | 0.726 |
| VILA-U (SigLIP-L/16-256) (Wu et al., 2025) | 73.3 | 10.0 | 11.2 | 1.80 | 3.43 | 0.489 |
| VILA-U (SigLIP-So/14-384) (Wu et al., 2025) | 78.0 | - | - | 1.25 | - | - |
| DualToken (SigLIP-L/16-256) | 79.8 | 20.8 | 21.4 | 1.06 | 27.12 | 0.693 |
| DualToken (SigLIP-So/14-384) | 82.0 | 21.5 | 21.6 | 0.25 | 28.69 | 0.744 |
| DualToken (SigLIP2-So/16-256) | 82.3 | 21.1 | 21.9 | 0.52 | 28.03 | 0.726 |

To disentangle the two conflicting objectives, we propose *interpreting visual appearance and visual semantics—required for visual generation and understanding—as distinct visual vocabularies*: a pixel codebook that captures low-level appearance features for generation, and a semantic codebook that encodes high-level semantic features essential for understanding. Specifically, inspired by the hierarchical structure of the human visual system (Groen et al., 2017), we partition the Vision Transformer (ViT) (Dosovitskiy et al., 2020) into shallow, middle, and deep stages based on the cosine similarity (Chen et al., 2025a) across layers and observe that shallow layers of a ViT predominantly capture low-level perceptual information—such as texture and color—making them suitable for reconstruction tasks, whereas high-level semantic representations emerge in the deeper layers (Chen et al., 2023b; 2025a). To fully exploit this inherent property of ViT, we utilize shallow-layer features for reconstruction and deep-layer features for semantic learning, thereby enabling the simultaneous derivation of both a pixel codebook and a semantic codebook within a unified tokenizer.

This hierarchical decoupling effectively resolves the conflict between the two objectives. As a result, our DualToken achieves the best semantic performance among established unified tokenizers (Wu et al., 2025; Zhao et al., 2025; Qu et al., 2024; Ma et al., 2025) while also attaining state-of-the-

art performance in reconstruction. Building upon this, we further demonstrate how a MLLM can effectively utilize the dual visual vocabularies to achieve unified vision understanding and generation.

Our analysis reveals three key findings: i) **Dual visual vocabularies resolve conflicts**: Decoupling visual appearance and visual semantics with separate visual vocabularies mitigates the conflict between reconstruction and semantic objectives. Our tokenizer achieves state-of-the-art performance in both sides, using only 10% of the pretraining data required by VILA-U; ii) **DualToken is better than combining dual encoders**: We observe that DualToken, as a unified architecture, outperforms the direct combination of two heterogeneous visual encoders, demonstrating both simplicity and effectiveness; iii) **Dual-token promote each other**: On one hand, visual appearance tokens (pixel tokens) are not only used for generation but also contribute fine-grained low-level features that enhance visual understanding. On the other hand, visual semantic tokens—beyond their role in understanding tasks—act as positive supervision during autoregressive generation, leading to more semantically aligned image outputs compared to generating pixel tokens alone.

## 2 RELATED WORKS

**Unified Multimodal Models**   A classic strategy for integrating visual understanding and generation within a single MLLM is to externally connect an LLM with a Diffusion Model (Sun et al., 2024; Dong et al., 2024; Pan et al., 2025; Chen et al., 2025b). However, pure AR architectures offer a more elegant, fully end-to-end solution by unifying both tasks within the same autoregressive framework. Representative works like Chameleon (Yu et al., 2023a; Team, 2024) and Emu3 (Wang et al., 2024b), have demonstrated the feasibility of jointly modeling vision and language through a unified next-token prediction objective. Specifically, visual inputs are first tokenized into visual tokens. These visual tokens are then interleaved with text tokens to construct a multimodal sequence. However, these pure AR architectures introduce generative capabilities at the cost of considerably weaker visual understanding. An empirical explanation for this (Wu et al., 2025; Xie et al., 2025b) is that their vision tokenizers are trained solely for reconstruction and thus primarily captures low-level visual details for generation rather than the high-level semantics required for vision–language understanding.

A straightforward way to bypass such a conflict is to employ two heterogeneous vision encoders (Wu et al., 2024a; Chen et al., 2025c; Deng et al., 2025b): a semantic tokenizer (*e.g.* CLIP) for understanding and a reconstruction-based tokenizer (*e.g.* VQ-VAE) for generation. Yet this design inevitably adds extra modules and structural complexity, making understanding and generation two loosely coupled systems with distinct pathways rather than a truly unified model. In contrast, the text modality relies on a single tokenizer (*e.g.*, BPE) (Sennrich et al., 2015) that discretizes text into a unified token space. This ensures a consistent input–output space: the input tokens that provide signals for understanding and the output tokens produced during generation share the same vocabulary. This unified design allows LLMs to seamlessly integrate text understanding and generation within the next-token prediction paradigm, thereby supporting broad generalization across diverse linguistic tasks. Therefore, the visual modality urgently requires a tokenizer that, like text tokenizers, can support both understanding and generation within a unified, coherent token space.

**Unified Visual Tokenizers**   Recent research has actively explored solutions in this direction. VILA-U (Wu et al., 2025) and MUSE-VL (Xie et al., 2025b) strive to build a unified tokenizer by jointly training on both reconstruction and semantic objectives. However, due to the inherent disparity between semantic and texture features, they struggle to strike an optimal balance between the two objectives, resulting in subpar performance in both tasks. As discussed in FQGAN (Bai et al., 2024), decomposing the codebook in a divide-and-conquer manner may offer a more fundamental solution to this conflict. TokenFlow (Qu et al., 2024) employs separate codebooks with a shared-mapping mechanism. However, key differences set our approach apart: (i) TokenFlow relies on distinct vision towers to extract semantic and low-level features, rather than leveraging a unified architecture; (ii) the shared IDs obtained through the shared-mapping mechanism may not be the optimal matches for either semantics or texture, potentially introducing additional losses in both domains.

## 3 METHOD

This section introduces the design of our unified tokenizer and explains how its dual visual codebooks are utilized within the next-token prediction paradigm for multimodal understanding and generation.

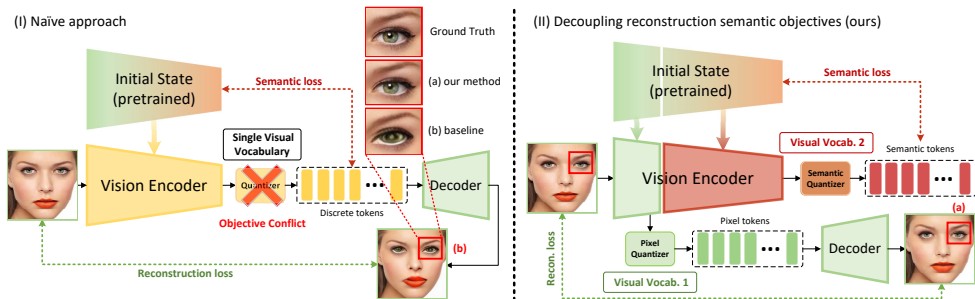

Figure 2: **Comparing the design of a naive (Left) and our decoupled approach (Right).** Naively combining reconstruction and semantic loss with a single visual vocabulary leads to distorted reconstruction and degraded semantic performance. We decouple the two objectives through a hierarchy approach, where reconstruction loss is applied to supervise the shallow layers, while semantic supervision is applied to the deep layers. This enhances both reconstruction fidelity and semantic quality. Consequently, we derive two complementary visual vocabularies: a pixel codebook for low-level visual appearance, and a semantic codebook for high-level visual semantics.

## 3.1 MOTIVATION AND VERIFICATION

As discussed in Qu et al. (2024), CLIP encoders cluster images by semantic similarity, whereas VQVAE-based encoders group images by low-level attributes such as color and texture. This suggests that encoders trained for reconstruction primarily capture low-level visual appearance, while those trained with text alignment excel at capturing high-level semantics. We argue that this difference in representation space is a key factor underlying downstream MLLM performance. Yet such a claim has not been formally validated before.

Table 2: **Downstream visual understanding performance with different vision encoders within the LLaVA-1.5 framework.** The *CLIP-based* encoder corresponds to the siglip-so400m-14-384 model (Alabdulmohsin et al., 2023), whereas *CLIP-based (recon.)* denotes an encoder with the same architecture but trained solely for reconstruction from scratch, controlling for factors like model size and architecture. For the *VQVAE-based* encoder, we adopt SBER-MoVQGAN-270M, a well-established reconstruction model.

| Vision Encoder Type | MMB↑ | MME↑ | SEED↑ | VQAv2↑ | Zero-Shot↑ | rFID↓ |
|---|---|---|---|---|---|---|
| CLIP-based | 61.8 | 1492.9 | 58.4 | 78.5 | 83.2 | ✗ |
| CLIP-based (recon.) | 36.2 | 822.4 | 30.6 | 47.5 | ✗ | 0.96 |
| VQVAE-based | 35.8 | 792.0 | 34.1 | 45.2 | ✗ | 0.68 |

To validate this viewpoint, we started by a preliminary experiment following the LLaVA-1.5 pipeline (Liu et al., 2024b). In Table.2, compared to the original SigLIP model, encoders trained with reconstruction objective exhibit a significant drop in downstream MLLM vision-language understanding performance, validating that high-level semantic features are more critical for visual reasoning in MLLMs than low-level perceptual features. However, to achieve both visual understanding and generation within a single MLLM, it is essential to decode the visual tokens back into pixel space as accurately as possible. However, since the SigLIP encoder focuses on high-level semantic information rather than texture details, simply discretizing its features and training a decoder without tuning the encoder results in poor image reconstruction quality. Therefore, proposing a unified tokenizer is crucial to enable high-quality visual understanding and generation within a singe MLLM.

## 3.2 UNIFIED VISION TOKENIZER WITH DUAL CODEBOOKS

To build a unified tokenizer, we started with the simplest approach, where we directly combine the reconstruction loss and semantic loss to optimize the entire vision tower and use a single visual vocabulary to tokenize its feature, similar to VILA-U (Wu et al., 2025). Specifically, as illustrated in Fig.2 (left), we initialize the vision encoder with pretrained weights from SigLIP (Zhai et al., 2023) to ensure strong text-image alignment. Then the semantic loss is computed between the deeper-layer features of the model and its initial state to constrain the model from losing its semantic capability.

However, as shown in Table.3 (a), this straightforward approach leads to a clear conflict between the two objectives. On one hand, although the semantic loss is applied to preserve the model's original semantic representation capabilities, achieving this objective proves difficult, as semantic performance metrics show a significant decline compared to the original model, reflecting the disruption caused by

Table 3: **DualToken resolves the conflict between reconstruction and semantic objectives.** Directly combining the two objectives leads to a drastic decline in reconstruction performance (a vs. b), while incorporating reconstruction and semantic losses hierarchically results in better reconstruction performance (a vs. d). We highlight our method in the last row. We adopt the pretrained weights from the siglip-so400m-patch14-384 in this experiment.

| # Exp. | Learning Objective (layer) | Feature Type | Zero-Shot Acc.↑ | Reconstruction | | |
|--------|---------------------------|--------------|-----------------|------|------|------|
| | | | | rFID↓ | PSNR↑ | SSIM↑ |
| *Initial State* | | Continuous | 83.2 | ✗ | ✗ | ✗ |
| *Initial State (quantized)* | | Discrete | 82.4 | ✗ | ✗ | ✗ |
| (a) | Recon. (26) + Sem. (26) | Discrete | 72.3 | 3.86 | 12.64 | 0.574 |
| (b) | Recon. (26) | Discrete | ✗ | 0.27 | 27.88 | 0.722 |
| (c) | Recon. (6) | Discrete | ✗ | 0.29 | 28.12 | 0.745 |
| (d) | Recon. (6) + Sem. (26) | Discrete | 82.0 | 0.25 | 28.69 | 0.744 |

the reconstruction training objective on semantic capabilities. On the other hand, as shown in the cropped region of Fig.2, the model also struggles to achieve satisfactory reconstruction quality, often producing distorted and blurry images.

To resolve this conflict, we begin by analyzing the intrinsic properties of the SigLIP encoder. Specifically, we divide the ViT into shallow, middle, and deep layers based on the cosine similarity of features across layers, as shown in Fig.3 (left). Guided by this partition, we extract features from the shallow and deep layer to perform clustering on the image representations. As shown in Fig.3 (right), we observe that features from the shallow layer tend to cluster images based on low-level attributes such as color and texture, whereas features from the deep layer form clusters according to semantics. This suggests that shallow SigLIP features capture fine-grained perceptual details, while deeper layers encode high-level semantics, aligning naturally with the respective requirements of downstream visual generation and understanding tasks.

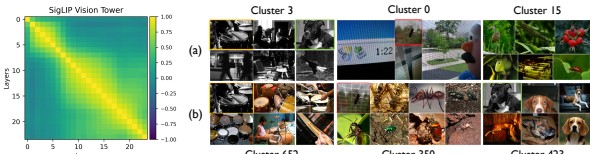

Figure 3: **(Left)** Partitioning of the SigLIP encoder (Zhai et al., 2023) based on the cosine similarity of features across layers. Distinct bright square regions are observed in the ranges of layers 1–7 and 8–17, indicating strong intra-group similarity within each interval; the remaining layers are treated as deep layers. **(Right)** Visualization of image clusters derived from features of (a) the 6th layer and (b) the 26th layer of SigLIP. Features from deep layers cluster images based on semantic content, whereas features from shallow layers form clusters based on low-level cues such as color and texture. For example, images in cluster 0 exhibit similar grid-like textures (e.g., window screens or monitor meshes). Implementation details of the clustering process are provided in Appendix.F.

Motivated by this, we introduce a hierarchical approach to decouple the learning of the reconstruction and semantic objectives. Specifically, as shown in Fig.2 (right), reconstruction loss is applied to supervise the shallow layers (1-6) of the vision tower, while semantic loss is applied to the deep 26-th layer (Please refer to Appendix.B for the selection of the reconstruction layer). Features from the shallow and deep layers are discretized separately via residual vector quantization (Lee et al., 2022), resulting in low-level and high-level visual vocabularies, referred to as the pixel codebook and the semantic codebook, respectively. To ensure the encoder outputs align closely with the codebook entries, we utilize a Vector Quantization (VQ) commitment loss, which is defined as

$$\mathcal{L}_c = \|z - \text{quantize}(z)\|_2^2 \tag{1}$$

Consequently, the total loss is formulated as a weighted sum of reconstruction loss, semantic loss, and VQ commitment loss

$$\mathcal{L}_{total} = \lambda_1 \cdot \mathcal{L}_{recon} + \lambda_2 \cdot \mathcal{L}_{sem} + \lambda_3 \cdot (\mathcal{L}_{c1} + \mathcal{L}_{c1}) \tag{2}$$

where the reconstruction loss is the combination of pixel-wise $L2$ loss (Dosovitskiy & Brox, 2016), LPIPS loss (Zhang et al., 2018) and adversarial loss (Isola et al., 2017) for reconstructing an input image

$$\mathcal{L}_{recon} = \|\hat{x} - x\|_2^2 + \lambda_p \mathcal{L}_{\text{LPIPS}}(\hat{x}, x) + \lambda_g \mathcal{L}_{\text{G}}(\hat{x}) \tag{3}$$

while the semantic loss is computed as the distance between the model's final-layer feature $F$ and its initial value $F_0$

$$\mathcal{L}_{sem} = -\cos(F, F_0) + \|F - F_0\|_2^2 \tag{4}$$

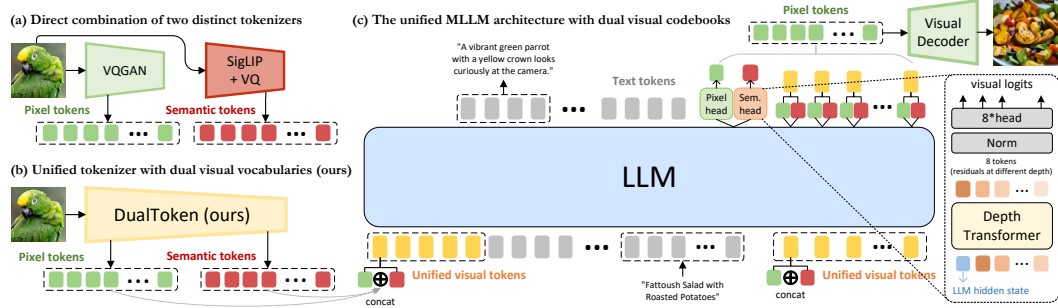

Figure 4: **(a) Direct combination of two heterogeneous tokenizer.** Baseline method (Huang et al., 2025) that directly uses VQGAN and CLIP-based encoder to separately acquire high-level (semantic) and low-level (pixel) visual codebooks. **(b) Our unified tokenizer with dual codebook.** We decoupling high-level and low-level visual codebooks within a unified vision tokenizer. The image is converted into low-level visual appearance tokens (green) and text-aligned semantic tokens (red). **(c) Architecture for unifying generation and understanding task.** In image generation task, the generated low-level tokens are decoded by the visual decoder to reconstruct the visual content.

where $\cos(\cdot)$ denotes cosine similarity. Interestingly, as shown in Table.3 (d), even without adding an additional contrastive learning phase, but solely by applying a simple constraint on the semantic representation, incorporating a reconstruction objective within our hierarchical learning strategy causes minimal damage to the model's semantic capability. Moreover, as shown in Table.3 (b)(c)(d), compared to training solely for reconstruction, incorporating semantic supervision in the deeper layers does not degrade reconstruction performance in the shallow layers, effectively resolving the conflict between semantic and reconstruction objectives.

## 3.3 UNIFYING UNDERSTANDING AND GENERATION

In this section, we demonstrate how to integrate the dual visual codebooks of DualToken within a unified MLLM. As illustrated in Fig.4 (c), to model both textual and visual content within the autoregressive paradigm of LLMs, the pixel and semantic visual tokens are first passed through a 2-layer MLP projector to align their dimensions with the LLM backbone. These tokens are then concatenated **along the embedding dimension** (which does not increase the sequence length) to form unified visual tokens. Next, the unified visual tokens are concatenated with text tokens to construct a multimodal token sequence. The model is then trained in an autoregressive manner to predict the next token across both visual and textual content.

For simplicity, we define the language vocabulary of our MLLM as a finite set $\mathcal{X} = \{x_1, x_2, ..., x_{n_1}\}$, while the low-level and high-level visual vocabulary as $\mathcal{Y} = \{y_1, y_2, ..., y_{n_2}\}$ and $\mathcal{Z} = \{z_1, z_2, ..., z_{n_3}\}$, where $n_1$, $n_2$, and $n_3$ represent the vocabulary sizes for language tokens, low-level visual tokens, and high-level visual tokens, respectively.

For visual tokens, since residual quantization introduces a depth-stacked structure of codes at each visual position $p$, we implement our visual heads based on the depth transformer from RQ-VAE (Lee et al., 2022). As shown in Fig.4, the semantic tokens and pixel tokens are processed by independent visual heads—the pixel head and the semantic head. Both heads share the same structure, comprising three layers of depth transformers and corresponding classification head for each depth.

Given the LLM hidden state $h_p$ for visual tokens at position $p$, our depth transformer autoregressively predicts D residual tokens $(r_{p1}, r_{p2}, ..., r_{pD})$. For $d > 1$, the input to the depth transformer at depth d, denoted as $I_{pd}$, is defined as the sum of the token embeddings of up to depth $d - 1$

$$I_{pd} = \sum_{d'=1}^{d-1} \mathbf{e}(r_{pd'}), \tag{5}$$

where $r \in \mathcal{Y}$ for the pixel head and $r \in \mathcal{Z}$ for the semantic head. The initial input at depth 1 is given by $I_{p1} = h_p$. This formulation ensures that the depth transformer incrementally refines the predicted feature representation by leveraging previous estimations up to depth $d - 1$. Consequently, the overall

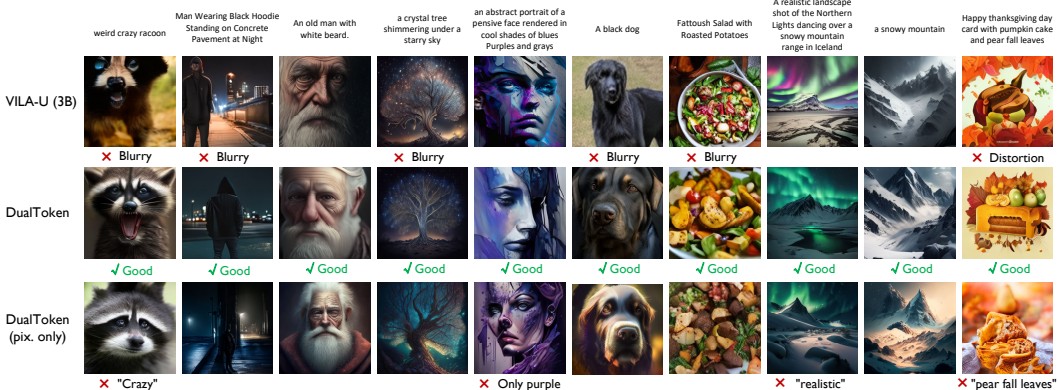

Figure 5: Qualitative results on visual generation.

negative log-likelihood loss for the entire multimodal sequence of length $N$ is defined, if a text token appears at position $i$, as

$$\mathcal{L}_{\text{NTP}} = -\sum_{i=1}^{N} \mathcal{P}_i,, \text{ where } \mathcal{P}_i = \log P\left(x_i | x_{<i}\right) \tag{6}$$

and if visual tokens appears at position $i$, as

$$\mathcal{P}_i = \sum_{d=1}^{D} \left[\log P\left(y_{id} | y_{i,<d}\right) + \log P\left(z_{id} | z_{i,<d}\right)\right] \tag{7}$$

## 4 EXPERIMENTS

### 4.1 VISION TOKENIZER

**Experimental Setup**  We trained two versions of our vision tokenizers at $256 \times 256$ and $384 \times 384$ resolutions. For fair comparison with VILA-U, we adopted the same quantization strategies and pretrained weights (SigLIP-L/16-256 and SigLIP-so/14-384), yielding 256 / 729 tokens with residual depths $D = 4$ / $D = 8$ (whereas VILA-U uses $D = 4$ / $D = 16$). To test stronger backbones, we further trained on SigLIP2-so/16-256 with $D = 8$, and show that our method generalizes to other backbones in Appendix. B. All models were trained on ImageNet-1K (Deng et al., 2009), CC12M (Changpinyo et al., 2021), and 50M images from LAION-400M (Schuhmann et al., 2021).

**Reconstruction**  We measured reconstruction FID (rFID), PSNR, and SSIM on the ImageNet-1K (val). As shown in Table.1, our DualToken achieves the highest structural similarity and the lowest rFID among various state-of-the-art dedicated methods, including Open-MAGVIT2 (Luo et al., 2024) and SBER-MoVQGAN (SberBank, 2023). This demonstrates that our method effectively mitigates the structural distortion and blurriness issues encountered by VILA-U during reconstruction.

**Semantic Metrics**  For semantic metrics, we report the Top-1 accuracy for zero-shot classification on ImageNet-1K (val), along with text-to-image and image-to-text retrieval performance (R@1) on Flickr8K. As shown in Table.1, our DualToken significantly outperforms VILA-U and the latest concurrent work, UniTok, while also surpassing dedicated models like CLIP-L-14-336 in zero-shot image classification and achieves performance on par with the state-of-the-art SigLIP models.

**Downstream Performance within LLaVA-1.5**  Before formally introducing the performance of our unified model, we first conducted a controlled experiment to validate the effectiveness of our vision tokenizer in downstream MLLM understanding tasks within the LLaVA-1.5 (Liu et al., 2024b) framework. Specifically, we replace the vision encoder of LLaVA-1.5 with DualToken, while strictly adhering to its training data and using LLaMA-2-7B (Touvron et al., 2023) as the foundational LLM. As shown in Table.4 (a)(b)(d), our DualToken, as a discrete unified vision tokenizer, outperforms VILA-U and even surpasses the original continuous SigLIP model.

Table 4: **Controlled comparison across ten visual understanding benchmarks**. We evaluate different vision encoders/tokenizers, including siglip-large-16-256, VILA-U, and DualToken within the LLaVA-1.5 framework. MMB refers to MMBench-dev (Liu et al., 2023), OCRB to OCRBench (Liu et al., 2024d), and TVQA to TextVQA (Singh et al., 2019). The MME (Fu et al., 2024) score is normalized based on its total score. *Sem.+Pix.* is the original setting of DualToken, where semantic and pixel tokens are concated along embedding dimension to serve as visual input. *Sem. only* means only the semantic tokens are fed as visual input.

| | Vision Encoder | Res. | MMB | MME | SEED | VQAv2 | MMVet | AI2D | MMMU | POPE | OCRB | TVQA | AVG. |
|---|---|---|---|---|---|---|---|---|---|---|---|---|---|
| (a) | siglip-large-16-256 | 256 | 60.9 | 62.9 | 56.4 | 78.2 | 34.5 | 53.5 | 30.8 | 80.3 | 26.3 | 44.3 | 52.8 |
| (b) | VILA-U | 256 | 55.3(-5.6) | 53.8(-9.1) | 51.2(-5.6) | 73.1(-5.1) | 24.9(-9.6) | 49.4(-4.1) | 28.4(-2.4) | 78.2(-2.1) | 23.8(-2.5) | 42.8(-1.5) | 48.1(-4.7) |
| (c) | DualToken (sem. only) | 256 | 59.8(-1.1) | 63.0(+0.1) | 56.2(-0.2) | 77.6(-0.6) | 34.0(-0.5) | 53.7(+0.2) | 30.3(-0.5) | 79.4(-0.9) | 24.6(-1.7) | 43.2(-1.1) | 52.2(-0.6) |
| (d) | DualToken (sem.+ pix.) | 256 | 61.3(+0.4) | 64.6(+1.7) | 57.2(+0.8) | 77.0(-1.2) | 34.6(+0.1) | 55.9(+2.4) | 30.2(-0.6) | 83.0(+2.7) | 29.2(+2.9) | 46.2(+1.9) | 53.9(+1.1) |

Table 5: Quantitative results on visual understanding and generation benchmarks.

| Type | Method | # Params | POPE | MMBench | SEED | MMMU | MMVet | MathVista | MME |
|---|---|---|---|---|---|---|---|---|---|
| | InstructBLIP (Dai et al., 2023) | 7B | - | 36.0 | 58.8 | 30.6 | 26.2 | 24.4 | 1137.1 |
| | LLaVA-Phi (Zhu et al., 2024) | 2.7B | 85.0 | 59.8 | - | - | 28.9 | - | 1335.1 |
| | LLaVA-1.5 (Liu et al., 2024b) | 7B | 85.9 | 64.3 | 58.6 | 35.4 | 31.1 | 27.4 | 1510.7 |
| *Und.* | LLaVA-NeXT (Liu et al., 2024c) | 7B | 86.5 | 67.4 | 70.2 | 35.8 | 43.9 | 34.6 | 1519.0 |
| | LLaVA-NeXT (Liu et al., 2024c) | 34B | 87.7 | 79.3 | 75.9 | 51.1 | 57.4 | 46.5 | 1631.0 |
| | ShareGPT4V (Chen et al., 2024b) | 7B | - | 68.8 | 69.7 | 37.2 | 37.6 | 26.5 | 1567.4 |
| | VILA (Lin et al., 2024a) | 7B | 85.5 | 68.9 | 61.1 | - | 34.9 | - | 1533.0 |
| | BAGAL (Deng et al., 2025a) | 14B | - | 85.0 | - | 55.3 | 67.2 | 73.1 | 1687.0 |
| | Chameleon (Team, 2024) | 7B | - | 31.1 | - | 22.4 | 8.3 | - | - |
| | Emu3 (Wang et al., 2024b) | 8B | 85.2 | 58.5 | 68.2 | 31.6 | - | - | - |
| | Show-o (Xie et al., 2024) | 1.5B | 73.8 | - | - | 25.1 | - | - | 948.4 |
| | Janus (Wu et al., 2024a) | 1.5B | 87.0 | 69.4 | 63.7 | 30.5 | 34.3 | - | 1338.0 |
| | Liquid (Wu et al., 2024b) | 7B | 83.2 | - | - | - | - | - | 1448.0 |
| | MUSE-VL (256px) (Xie et al., 2025b) | 7B | - | 72.1 | 69.1 | 39.7 | - | 51.3 | 1480.9 |
| *Uni.* | TokenFlow (384px) (Qu et al., 2024) | 13B | 86.8 | 68.9 | 68.7 | 38.7 | 40.7 | - | 1545.9 |
| | UniTok (256px) (Ma et al., 2025) | 7B | 83.2 | 61.1 | - | - | 33.9 | - | 1448.0 |
| | UniToken (384px) (Jiao et al., 2025) | 7B | - | 71.1 | 69.9 | 32.8 | - | 38.5 | - |
| | Show-o2 (432px) (Xie et al., 2025a) | 1.5B | - | 67.4 | 65.6 | 37.1 | - | - | 1450.9 |
| | Show-o2 (432px) (Xie et al., 2025a) | 7B | - | 79.3 | 69.8 | 48.9 | - | - | 1620.5 |
| | VILA-U (Wu et al., 2025) | 7B | 85.8 | - | 59.0 | - | 33.5 | - | 1401.8 |
| | DualToken-3B (256px) | 3B | 86.0 | 70.9 | 70.2 | 38.6 | 32.5 | 46.5 | 1489.2 |
| | DualToken-3B (384px) | 3B | 88.1 | 76.2 | 72.2 | 40.3 | 40.2 | 49.2 | 1588.4 |
| | DualToken-7B (256px) | 7B | 88.6 | 74.9 | 71.8 | 45.8 | 40.5 | 55.8 | 1502.7 |
| | DualToken-7B (384px) | 7B | 89.4 | 80.0 | 72.5 | 47.4 | 44.3 | 57.6 | 1625.0 |

(a) Evaluation on multimodal understanding benchmarks.

| Type | Method | Architecture | Count↑ | Differ↑ | Compare↑ | Logical↑ Negate | Logical↑ Universal | Overall↑ |
|---|---|---|---|---|---|---|---|---|
| | SD-XL (Podell et al., 2023) | Diffusion | 0.71 | 0.73 | 0.69 | 0.50 | 0.66 | 0.63 |
| *Gen.* | Midjourney v6 (Midjourney, 2024) | Diffusion | 0.78 | 0.78 | 0.79 | 0.50 | 0.76 | 0.69 |
| | DALL-E 3 (Betker et al., 2023) | Diffusion | 0.82 | 0.78 | 0.82 | 0.48 | 0.80 | 0.70 |
| | Show-o (Xie et al., 2024) | Discrete Diff. | 0.70 | 0.62 | 0.71 | 0.51 | 0.65 | 0.60 |
| | ILLUME (Wang et al., 2024a) | AR+Diff. | 0.66 | 0.68 | 0.67 | 0.49 | 0.63 | 0.60 |
| | LWM (Liu et al., 2024a) | Autoregressive | 0.59 | 0.58 | 0.54 | 0.49 | 0.52 | 0.53 |
| | Liquid (Wu et al., 2024b) | Autoregressive | 0.76 | 0.73 | 0.74 | 0.46 | 0.74 | 0.65 |
| *Uni.* | UniTok (Ma et al., 2025) | Autoregressive | 0.76 | 0.76 | 0.79 | 0.46 | 0.73 | 0.67 |
| | VILA-U (Wu et al., 2025) | Autoregressive | 0.70 | 0.71 | 0.74 | 0.53 | 0.66 | 0.64 |
| | VILA-U 3B (256) | Autoregressive | 0.68 | 0.66 | 0.70 | 0.49 | 0.64 | 0.60 |
| | DualToken-3B (256) | Autoregressive | 0.76 | 0.76 | 0.78 | 0.50 | 0.72 | 0.68 |
| | DualToken-3B (pix. only) | Autoregressive | 0.59 | 0.59 | 0.59 | 0.47 | 0.59 | 0.55 |

(b) VQAScores on *advanced* prompts of GenAI-Bench (Lin et al., 2024b)

## 4.2 UNIFIED MODEL FOR GENERATION AND UNDERSTANDING

Building on the unified tokenizers, we further verified its potential within a unified AR framework based on Qwen-2.5-3B (Yang et al., 2024). Our training process consists of four stages: (1) Freeze the LLM and pretrain on image-caption data, training only the visual projector for multimodal alignment. (2) Unfreeze the LLM and fine-tune on visual understanding data to enhance comprehension. (3) Freeze the LLM and train only the visual heads on text-to-image data. (4) Unfreeze all components and perform joint training on a mixture of understanding, generation, and interleaved datasets.

To ensure a fair comparison with VILA-U (Wu et al., 2025), we additionally provide a reproduced version of VILA-U using Qwen-2.5-3B as the language backbone, trained with the same dataset and procedure as our method. We evaluate our model against widely used vision-language understanding benchmarks, including VQAv2 (Goyal et al., 2017), POPE (Li et al., 2023b), MME (Fu et al., 2024), SEED-IMG (Li et al., 2023a), MMBench (Liu et al., 2023), and MM-Vet (Yu et al., 2023b).

As shown in Table.5, our DualToken (3B) demonstrates strong understanding performance compared to other unified models and surpasses dedicated understanding models like LLaVA-NeXT and ShareGPT4V (Chen et al., 2024b). Meanwhile, as illustrated in Fig. 5, thanks to the significantly improved reconstruction quality of DualToken, the generated images are rich in detail and structurally realistic, accurately capturing fine textures such as animal fur and other intricate patterns—effectively resolving the blurriness and distortions observed in VILA-U. What's more, the generated images exhibit remarkable alignment with the text, even for long and complex prompts. This is especially evident when compared with the *pix. only* method (which only predicts pixel tokens during image generation), as it often ignores important semantic content during generation—highlighting the crucial role that semantic tokens play in helping the model grasp the semantic structure of images throughout the generation process. Results on more generation benchmarks are presented in Appendix.E.

Beyond its impressive performance, we observed two interesting findings:

- *Pixel tokens enhance understanding.* As shown in Table.4 (a)(c)(d), we compared using only the semantic tokens (sem.), and a combination of semantic and pixel tokens (sem.+pcpt), concatenated along the embedding dimension to serve as visual input. Surprisingly, compared to using semantic tokens alone, jointly leveraging both semantic and pixel tokens leads to consistent improvements across various aspects, including general VQA (Liu et al., 2023; Fu et al., 2024), hallucination detection (Li et al., 2023b), and OCR-related benchmarks (Singh et al., 2019; Liu et al., 2024d). Suggesting that the supplementation of high-frequency details by pixel tokens can compensate for the subtle semantic loss introduced by vector quantization.

- *Semantic tokens also helps to generate.* As shown in Fig.5 and Table.5 (b), incorporating semantic tokens into the model's autoregressive generation process leads to more semantically aligned image generation compared to using visual appearance tokens alone. This indicates that visual semantic tokens—beyond their role in understanding tasks—can also assist the model in grasping the semantic composition of images, thereby producing outputs that better align with the intended semantics. This is also clearly reflected in the model's performance on GenAI-Bench.

**DualToken versus dual-encoder.** Recently, some studies have adopted dual-encoder designs to obtain visual representations (Huang et al., 2025). Specifically, a VQVAE-based pixel encoder and a CLIP-based semantic encoder. To address a fundamental question—why is it necessary to obtain dual visual vocabularies within a unified tokenizer rather than simply combining existing specialized encoders? we conducted an experiment using the codebook from SBER-MoVQGAN as the low-level vocabulary and a VQ-processed SigLIP as the high-level vocabulary, as illustrated in Fig.4 (a).

As shown in Table.6, this straightforward approach leads to significantly inferior image generation performance (See Appendix.F.4 for implementation details). To explain this discrepancy, we visualize the feature spaces of DualToken's 6th and 26th layers, as well as those of MoVQGAN and SigLIP with UMAP (Fig.6).

Table 6: Results on the MJHQ-30K dataset (Li et al., 2024a).

| Method | Type | Res. | FID↓ |
|---|---|---|---|
| SD-XL (Podell et al., 2023) | Diffusion | 1024 | 9.55 |
| PixArt (Chen et al., 2023a) | Diffusion | 1024 | 6.14 |
| Playground (Li et al., 2024a) | Diffusion | 1024 | 4.48 |
| Liquid (Wu et al., 2024b) | Autoregressive | 512 | 5.47 |
| Janus (Wu et al., 2024a) | Autoregressive | 384 | 10.10 |
| LWM (Liu et al., 2024a) | Autoregressive | 256 | 17.77 |
| Show-o (Xie et al., 2024) | Discrete Diff. | 256 | 15.18 |
| VILA-U 7B (Wu et al., 2025) | Autoregressive | 256 | 12.81 |
| VILA-U 3B | Autoregressive | 256 | 15.12 |
| DualToken 3B | Autoregressive | 256 | 7.88 |
| *Dual Encoder* | Autoregressive | 256 | 17.55 |

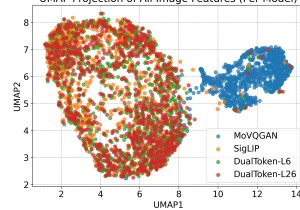

Figure 6: Visualized feature spaces on Imagenet-1k (val).

As shown, while DualToken's 6th and 26th layers yield features specialized for different purposes, they still share a largely overlapping representational space. In contrast, features from the two separate encoders (MoVQGAN and SigLIP) show significant divergence, forming clearly disjoint clusters. Therefore, we attribute the performance gap to the incompatibility of representational spaces between heterogeneous encoders. This mismatch imposes a burden on the downstream language model, which is forced to learn two entirely disjoint visual representation systems. This observation further highlights the simplicity and effectiveness of DualToken as a unified architectural solution.

## 5 CONCLUSION

This paper presents DualToken, which, to the best of our knowledge, is the first to demonstrate that a dual-codebook design—reconstructing from shallow layers while learning semantics from deep

layers—can effectively resolve the long-standing conflict between reconstruction and semantic objectives within a single visual tokenizer. Building upon DualToken, we develop a pure autoregressive (AR) unified model that achieves state-of-the-art performance in both understanding and generation among all existing discrete AR approaches. Orthogonal to concurrent works that focus on improving the VQ mechanism itself (Ma et al., 2025), our method emphasizes a hierarchical architectural design. Consequently, as more advanced VQ techniques emerge, our framework can naturally benefit from these improvements. We hope that DualToken offers a new perspective for designing unified visual tokenizers and sheds light on building a truly unified architecture for vision–language models.

ACKNOWLEDGMENTS

This work is partially supported by the National Natural Science Foundation of China under grant No. 62403389, the Provincial Natural Science Foundation of Zhejiang under grant No. QKWL25F0301, and the Zhejiang Key Laboratory of Low-Carbon Intelligent Synthetic Biology (2024ZY01025).

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

## A  LARGE LANGUAGE MODEL USAGE

In this paper, Large Language Models (LLMs) are used exclusively for grammatical error correction.

## B  LAYER SELECTION AND GENERALIZABILITY

As discussed in Sec. 3.2, we can partition the vision encoder into shallow and deep regions based on the cosine similarity between layers (see Fig. 3 and Appendix F.1). Empirically, selecting the last layer within the shallow region, typically corresponding to *the first quarter to third of layers*, yields the best results. As validated in the main text, our method successfully generalizes across multiple backbones, including SigLIP-L/16-256, SigLIP-SO/14-384, and SigLIP2-SO/16-256.

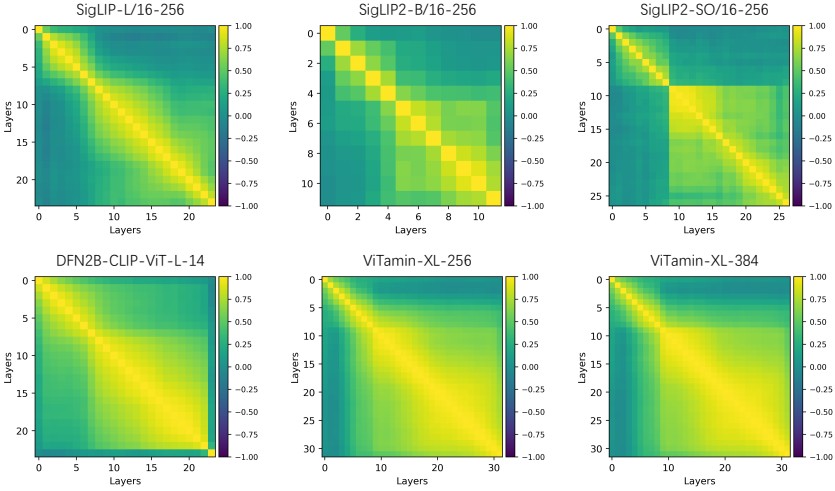

Figure 7: Additional visualizations of inter-layer cosine similarity across different backbones, revealing a consistent hierarchical pattern across architectures, model scales, and resolutions. A cohesive high-similarity region corresponding to the shallow layers can be clearly observed.

To further strengthen our claim, we include a comprehensive validation of the method's generalizability, covering six mainstream backbones—OpenAI's CLIP (Radford et al., 2021), Apple's DFN (Fang et al., 2023), BAAI's EVA (Fang et al., 2024), Google's SigLIP (Zhai et al., 2023), SigLIP2 (Tschannen et al., 2025), and the hybrid CNN–Transformer architecture ViTamin (Chen et al., 2024a)—under consistent settings (RVQ = 8, codebook size = 16,384, 2M training samples). Our findings are summarized as follows.

**(1) Ablations on model backbones and model size.** As shown in Table.7 and Table.8, across all tested backbones with varying types and total layer counts, selecting the *first quarter of layers* for reconstruction consistently yields the best reconstruction quality and semantic performance.

Table 7: Model backbone.    Table 8: Model size and total layer.    Table 9: Robustness on layers.

| Backbone | Layer recon./total | Zero-shot | rFID | Backbone | Layer recon./total | Zero-shot | rFID | Backbone | Layer recon./total | Zero-shot | rFID |
|---|---|---|---|---|---|---|---|---|---|---|---|
| | 6/24 | **79.2** | **0.84** | | 3/12 | **74.1** | **0.98** | DFN-B/16-224 | 3/12 | **74.1** | 0.98 |
| DFN-L/14-224 | 12/24 | 77.1 | 1.35 | DFN-B/16-224 | 6/12 | 71.4 | 1.58 | | 4/12 | 73.9 | **0.97** |
| | 18/24 | 72.2 | 3.25 | | 9/12 | 66.1 | 3.21 | | 5/24 | 79.0 | **0.82** |
| | 6/24 | **77.8** | **0.80** | | 6/24 | **79.2** | **0.84** | DFN-L/14-224 | 6/24 | **79.2** | 0.84 |
| EVA-02-L/14-224 | 12/24 | 75.2 | 1.16 | DFN-L/14-224 | 12/24 | 77.1 | 1.35 | | 7/24 | 78.9 | 0.84 |
| | 18/24 | 69.9 | 3.22 | | 18/24 | 72.2 | 3.25 | | 6/32 | **81.1** | 0.73 |
| | 6/24 | **73.2** | **0.87** | | 8/32 | **81.0** | **0.73** | DFN-H/14-224 | 8/32 | 81.0 | 0.73 |
| CLIP-L/14-224 | 12/24 | 70.8 | 1.80 | DFN-H/14-224 | 16/32 | 79.5 | 1.28 | | 10/32 | 80.7 | **0.72** |
| | 18/24 | 65.5 | 3.58 | | 24/32 | 74.3 | 2.78 | | 5/24 | 80.4 | 0.74 |
| | 6/24 | **78.8** | **0.78** | | 6/24 | **80.4** | **0.72** | SigLIP2-L/16-256 | 6/24 | **80.4** | **0.72** |
| SigLIP-L/16-256 | 12/24 | 76.3 | 1.27 | SigLIP2-L/16-256 | 12/24 | 77.6 | 1.09 | | 7/24 | 80.2 | 0.75 |
| | 18/24 | 72.9 | 2.61 | | 18/24 | 72.9 | 2.93 | | 5/27 | 81.0 | 0.70 |
| | 6/24 | **80.4** | **0.72** | | 7/27 | **81.2** | **0.69** | | 6/27 | 81.2 | **0.66** |
| SigLIP2-L/16-256 | 12/24 | 77.6 | 1.09 | SigLIP2-SO/16-256 | 14/27 | 78.4 | 1.08 | SigLIP2-SO/16-256 | 7/27 | 81.2 | 0.69 |
| | 18/24 | 72.9 | 2.93 | | 21/27 | 73.7 | 3.02 | | 8/27 | **81.3** | 0.67 |
| | 6/24 | **80.0** | **0.39** | | | | | | 9/27 | 81.0 | 0.72 |
| ViTamin-XL-384 | 12/24 | 78.6 | 0.88 | | | | | | | | |
| | 18/24 | 73.8 | 2.25 | | | | | | | | |

**(2) Robustness to specific layer choice.** As shown in Table. 9, our method remains stable as long as the reconstruction layers lie roughly within the first quarter of the network. Minor shifts ($\pm 1$–2 layers) cause negligible changes, confirming its robustness and broad applicability.

These findings indicate that for a new ViT architecture, one can confidently select the first quarter of layers for reconstruction to achieve optimal results. Moreover, As shown in Fig. 7, we also provide more visualizations of inter-layer cosine similarity across backbones, revealing a consistent hierarchical pattern divisible into shallow and deep layers (also supported by prior study (Chen et al., 2025a)), further supporting the universality of our method.

## C DISCUSSION AND COMPARISON WITH UNITOK

Since UniTok adopts a more advanced visual backbone, decoder, and discriminator architecture, we conduct a fair comparison by re-training our DualToken under the same encoder and decoder settings used in UniTok, *i.e.*, choosing ViTamin-L/16, a hybrid architecture of CNN and transformer, to instantiate DualToken. Under this setup, DualToken achieves stronger semantic performance and competitive reconstruction quality, as evidenced by the comparison between (a) and (b) in Table 10.

Table 10: Comparison with UniTok.

| Tokenizer | rFID ↓ | Zero-Shot Acc ↑ |
|---|---|---|
| (a) UniTok | 0.38 | 78.6 |
| (b) DualToken (RVQ) | 0.39 | 80.3 |
| (c) DualToken (MCQ) | 0.25 | 82.2 |

Furthermore, *DualToken and UniTok are complementary*. Specifically, by replacing our original RVQ quantizer with UniTok's proposed **MCQ**, we observe consistent improvements in both reconstruction fidelity and zero-shot classification, as evidenced by the comparison between (b) and (c) in Table 10.

These results suggest that future work may benefit from **integrating our dual visual vocabulary** formulation with more advanced quantizers such as MCQ.

## D COMPUTATIONAL ANALYSIS

Introducing two codebooks **DOES NOT** significantly increase the computational overhead, demonstrated by two aspects: **parameter count** and **memory usage with inference latency**.

### D.1 PARAMETER COUNT

The ONLY additional parameters arise from 3 components:

- The MLP projector's dimension changes from (1024→2048→2048) to (2048→2048→2048), which adds **2.1M** parameters.
- An additional visual head: **258M** parameters.
- An additional VQEmbedding layer: **16M** parameters.

Together, these account for only **8.93%** of the total parameters compared to the LLM backbone (3B). When scaling to larger backbones (e.g., 7B), the relative impact becomes even more negligible.

### D.2 MEMORY USAGE AND INFERENCE LATENCY

Since our dual tokens are concatenated along feature dimension rather than sequence dimension, and the input dimension to the LLM remains unchanged, **no new pathway is introduced to the LLM**, and the computational cost of the LLM backbone remains strictly the same. The only increase stems from the components listed above.

Memory usage is measured under the same local batch size and device. FLOPs and inference time are averaged on T2I task (256px) over the MJHQ-30K dataset. Statistics for the VQA task have also been added to the paper.

Table 11: Memory Usage and Inference Latency

|  | Training Memory Usage | Inference Time Cost | Single Forward GFLOPs |
|---|---|---|---|
| single token | 73.8G | 11.42s | 328.98 |
| dual token | 78.2G | 12.97s | 337.20 |

## E    RESULTS ON MORE GENERATION BENCHMARKS

Following VILA-U, we report results on GenAI-Bench and MJHQ-30K in the main text. We now extend our evaluation to include GenEval (Ghosh et al., 2023) and WISE (Niu et al., 2025). The results demonstrate that DualToken achieves competitive performance across both benchmarks.

Table 12: Evaluation results on GenEval and WISE benchmarks.

| Model | GenEval (Overall) ↑ | WISE (Overall) ↑ |
|---|---|---|
| SDv1.5 (Rombach et al., 2022) | 0.43 | 0.32 |
| SDXL (Podell et al., 2023) | 0.55 | 0.43 |
| Chameleon-7B (Team, 2024) | 0.39 | - |
| EMU3-8B (Wang et al., 2024b) | 0.66 | 0.39 |
| Janus (Wu et al., 2024a) | 0.61 | 0.23 |
| Janus-Pro-7B (Chen et al., 2025c) | 0.80 | 0.35 |
| ILLUME-7B (Wang et al., 2024a) | 0.61 | - |
| TokenFlow-XL-14B (Qu et al., 2024) | 0.63 | - |
| Muse-VL-7B (Xie et al., 2025b) | 0.57 | - |
| UniToken-7B (Jiao et al., 2025) | 0.63 | - |
| Show-o2-1.5B (Xie et al., 2025a) | 0.73 | 0.35 |
| Show-o2-7B (Xie et al., 2025a) | 0.76 | 0.39 |
| VILA-U-7B (Wu et al., 2025) | - | 0.31 |
| DualToken-3B | 0.72 | 0.35 |
| DualToken-7B | 0.75 | 0.39 |

## F    IMPLEMENTATION DETAILS

### F.1    PARTITIONING OF THE SIGLIP ENCODER

We feed the ImageNet-1K (Deng et al., 2009) validation set into SigLIP-SO400M-Patch14-384 (Zhai et al., 2023). For each image, we extract the representations from all layers of the model, each with a shape of $729 \times 1152$. Then, we apply average pooling along the spatial dimension (the first axis) of each layer's representation, resulting in a 1152-dimensional vector per layer.

Specifically, for each image, we obtain feature vectors from 26 layers, and compute the pairwise cosine similarity between these layer-wise representations to construct a $26 \times 26$ cosine similarity matrix. To capture the overall similarity structure across layers in the model, we average the cosine similarity matrices across all images. The final similarity matrix $S^*$ is computed as:

$$S^* = \frac{1}{n} \sum_{i=1}^{n} S_i \tag{8}$$

where $S_i$ denotes the cosine similarity matrix for the $i - th$ image, and $n$ is the total number of images. $S^*$ thus represents the average inter-layer similarity across the dataset.

### F.2    IMAGE CLUSTERING

We extract intermediate representations from the 6th and 26th layers of SigLIP-SO400M-Patch14-384 (Zhai et al., 2023) for each image in the ImageNet-1K validation set (Deng et al., 2009). The original representation shape is $729 \times 1152$, and we apply average pooling along the spatial dimension to obtain a single 1152-dimensional feature vector per image. For both the 6th-layer and 26th-layer

features, we perform k-means clustering with 1000 cluster centers (Cluster 0 to Cluster 999). The cluster analysis reveals that shallow-layer features (from the 6th layer) tend to capture low-level visual attributes such as texture and color, while deep-layer features (from the 26th layer) predominantly encode high-level semantic content. The implementation code is provided in the *supplementary material*, and additional visualizations are presented in Fig. 8.

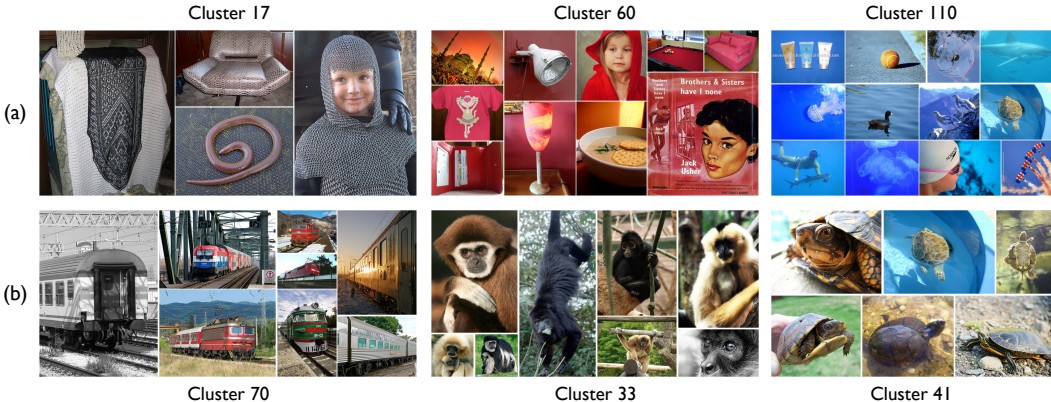

Figure 8: More visualizations of image clusters derived from features of (a) the 6th layer and (b) the 26th layer of SigLIP. Features from deep layers primarily cluster images based on high-level semantic content, whereas shallow-layer features tend to group images according to appearance-level cues such as color and texture. For instance, Cluster 17 contains images with similar scaly textures, while Clusters 60 and 110 predominantly group images by dominant colors (e.g., red or blue).

### F.3 UMAP Feature Space Visualization

We perform dimensionality reduction using UMAP to visualize the feature spaces from DualToken's 6th and 26th layers, as well as those from MoVQGAN and SigLIP. Specifically, we sample 1,000 images from the ImageNet-1K validation set and visualize the UMAP projections of their encoded features from each model. To ensure a fair comparison among the different visual models, all extracted features are first flattened and then uniformly processed via adaptive average pooling to maintain consistent dimensionality.

### F.4 Model Implementation Details

Our backbone model is built upon a decoder-only transformer architecture, and adopt Qwen2.5 (Yang et al., 2024) as our initialization due to its strong performance and public availability. The model uses RMSNorm Zhang & Sennrich (2019) for normalization. For visual inputs to the LLM, we apply a projector to map the visual tokens into the same embedding space as the LLM. When predicting image tokens, the output hidden states of the LLM are passed through two separate projectors to align with the dimension of the semantic visual head and the pixel visual head. Each projector consists of two linear layers with a GeLU activation in between. We use special tokens—<image_gen_start> and <image_gen_end>—to indicate the boundaries of the image to be generated.

For visual heads, since residual quantization introduces a depth-stacked structure of codes at each visual position $p$, we implement our visual heads based on the depth transformer from RQ-VAE (Lee et al., 2022). Unlike the original depth transformer, which employs a single head to predict logits across all depths, we introduce separate classification heads to compute the logits for residuals at each corresponding depth (Li et al., 2025a). As shown in Fig.4, the semantic tokens and pixel tokens are processed by independent visual heads—the pixel head and the semantic head. Both heads share the same structure, comprising three layers of depth transformers and corresponding classification head for each depth. Detailed training hyper-parameters are provided in Table 13.

**Implementation of the Dual Encoder Baseline** As described in Sec. 4.2 of the main paper, some concurrent works adopt *dual-encoder* designs to obtain visual representations (Huang et al., 2025), specifically combining a VQVAE-based pixel encoder with a CLIP-based semantic encoder.

Table 13: Training hyper-parameters.

| Settings | Visual Tokenizer | MLLM | | | | |
|---|---|---|---|---|---|---|
| | | Stage 1 | Stage 2 | Stage 3 | Stage 4-1 | Stage 4-2 |
| Learning Rate | 7.2e-5 | Projector 1e-3 | Projector 2e-5 LLM 2e-5 | Projector (Gen) 1e-4 Visual Heads 1e-4 | All Projectors 1e-5 Visual Heads 1e-5; LLM 1e-5 | |
| Batch Size | 64 | 64 | 256 | 128 | 512 | 256 |
| Optimizer | AdamW | AdamW | AdamW | AdamW | AdamW | AdamW |

This raises a natural question: *Beyond architectural elegance and simplicity, does learning dual visual codebooks within a unified visual tokenizer (ours) lead to better downstream performance in unified MLLMs compared to directly combining two heterogeneous encoders?*

Since these concurrent works adopt different training datasets and downstream architectures (e.g., external diffusion decoders (Rombach et al., 2022)), it is difficult to conduct a fair comparison in the context of downstream unified models. To isolate the effectiveness of the tokenization strategy itself—that is, dual tokens within a single unified tokenizer vs. dual visual tokenizers from separate encoders—we implemented both designs under the same unified architecture proposed in our work.

Specifically, we use SigLIP-L-Patch16-256 (Zhai et al., 2023) and SBER-MoVQGAN (SberBank, 2023) to build the semantic tokenizer and pixel tokenizer, respectively:

- *The semantic tokenizer* applies an RVQ quantizer (depth=4) to the penultimate layer of the frozen SigLIP-L-Patch16-256 encoder. The encoder is fully frozen, and only the codebook is updated using commitment loss, aiming to reconstruct the input semantic features as faithfully as possible.
- *The pixel tokenizer* is derived from a modified version of SBER-MoVQGAN-270M. To match the token length of SigLIP-L-Patch16-256, we added a downsampling and a upsampling modules to its encoder and decoder, adjusting the downsampling and upsampling rate from 8 to 16. Additionally, we replaced the original quantizer with a residual vector quantizer (RVQ) of depth 4 to ensure compatibility with our unified model architecture.

Apart from the different tokenizers used to provide pixel and semantic tokens, the rest of the architecture remains fully consistent with DualToken. Specifically, we concatenate pixel and semantic tokens along the embedding dimension to form the visual input, map them into the LLM embedding space via a projector, and use separate visual heads (pixel head & semantic head) for predictions. **To ensure fairness**, we standardized all other components except for the source of dual visual tokens:

- All components are kept identical, including image resolution, token length (16×16), RVQ depth ($D = 4$), embedding dimension, model architecture, and training data.
- Both tokenizers are trained on the same datasets as DualToken, as described in the main text.

## G  DATASETS

Our MLLM training process consists of four stages: (1) Freeze the LLM and pretrain on image-caption data, training only the visual projector for multimodal alignment. (2) Unfreeze the LLM and fine-tune on visual understanding data to enhance comprehension. (3) Freeze the LLM and train only the visual heads on text-to-image data. (4) Unfreeze all components and perform joint training on a mixture of understanding, generation, and interleaved datasets, enabling the model to acquire generative capabilities while maintaining strong understanding performance. We listed the data in Table. 14.

Table 14:  Training data list.

| Stage | Dataset |
|---|---|
| Visual Tokenizer | CC12M (Changpinyo et al., 2021), ImageNet-1K, 50M images from LAION-400M (Schuhmann et al., 2021) |
| MLLM Stage1 | DenseFusion-1M (Li et al., 2024b), DreamLIP (Zheng et al., 2024), InternVL-SA-1B-Caption (Chen et al., 2024c) |
| MLLM Stage2 | DocStruct4M (Hu et al., 2024), WebSight (Laurençon et al., 2024b), WuKong, 2M in house VQA data, pure text data |
| MLLM Stage3 | A filtered subset of ImageNet-21K, laion-aesthetics-12m, JourneyDB [a] (Sun et al., 2023) |
| MLLM Stage4 | In-house aesthetics data, OmniEdit (Wei et al., 2024), text2face, Cauldron (Laurençon et al., 2024a), Instruct-Pix2Pix (Brooks et al., 2022), Inhouse IFT data (Und.), OBELICS (Laurençon et al., 2023), pure text data |

[a]The text and image are reversed and used for image generation training.

