# OpenReview forum: "DualToken: Towards Unifying Visual Understanding and Generation with Dual Visual Vocabularies"
_ICLR.cc/2026/Conference — ICLR 2026 Poster_

### Official Review · Reviewer_kK8G · 2025-10-31

**Soundness:** 3
**Presentation:** 4
**Contribution:** 2
**Rating:** 6
**Confidence:** 5

**Summary:**

This paper tackles the challenge of unifying visual understanding and generation within the autoregressive framework of large language models. The authors propose DualToken, which disentangles high-level semantic and low-level visual representations through separate codebooks to reconcile their conflicting objectives. This design enables effective integration of reconstruction and semantic learning, achieving strong results on both understanding and generation tasks. Overall, the work presents a well-motivated direction toward unified multimodal representation learning.

**Strengths:**

1. The paper presents a clear and well-articulated motivation, with a logical flow from problem identification to solution design.

2. The evaluation covers a broad range of downstream tasks across both visual understanding and generation, demonstrating comprehensive empirical validation.

3. The proposed disentangled dual-codebook design is conceptually elegant and effectively resolves the conflict between reconstruction and semantic alignment.

**Weaknesses:**

1. Although the proposed disentangled dual-codebook design is effective, it lacks substantial architectural novelty. Moreover, the insights presented in this work have already appeared in recent vision tokenizer studies, such as UniTok.

2. The claim that “dual tokens promote each other” seems somewhat overstated. As shown in Table 4 (rows (c) and (d)), the pixel tokens do not consistently enhance performance on understanding tasks.

3. The reported downstream results are primarily based on experiments with a 3B model. Additional studies across different LLM scales are needed to validate the method’s generality and scalability.

4. The paper lacks a detailed ablation study on how the boundary between low-level and high-level layers in the vision encoder is determined, which would help clarify the effectiveness of the proposed disentanglement.

**Questions:**

N/A

---

> ### Author Response · Authors · 2025-11-21
>
> We sincerely thank the reviewer for the thoughtful and encouraging feedback. We are delighted that you found the paper:
> * presents a well-articulated motivation with a clear and logical flow,
> * is thoroughly validated through comprehensive experiments, and
> * proposes a method that is both *elegant* and *effective*.
>
> ---
>
> ### W1: Key Insight
>
> > ...The insights presented in this work have already appeared in recent vision tokenizer studies, such as UniTok.
>
> 1. The key insight of our work is that using *shallow layers* for pixel-level reconstruction yields better reconstruction quality while minimally affecting *deep-layer* semantic performance. To the best of our knowledge, we are the **FIRST** study to explicitly propose this *shallow-for-reconstruction, deep-for-semantics* principle for image tokenizers.
>
> 2. While *UniTok* also employs a *single tokenizer* for both reconstruction and semantics, it relies on the *same representation layer* for both objectives. In contrast, our method reveals that using *shallow layers* for pixel yields better reconstruction performance while causing less damage to *deep-layer* semantic performance. This design provides superior reconstruction quality and stronger semantic retention. Thus, although related in topic, our approach is **conceptually distinct and novel**.
>
> 3. DualToken and UniTok are in fact **COMPLEMENTARY** --- **We tackle the challenge from two orthogonal yet synergistic directions**: *UniTok* focuses on improving the *VQ mechanism* (via MCQ), while our method emphasizes *architectural disentanglement*. Incorporating our layer-wise design into UniTok further improves both reconstruction fidelity and zero-shot classification (Please refer to **Appendix C**).
>
> Moreover, we would like to kindly clarify that our work and UniTok are in fact *concurrent*.
>
> ### W2: Pixel Tokens Enhance Understanding
>
> Thank you for the comment. Performance on individual benchmarks may fluctuate due to the limited data scale of *LLaVA-1.5*. Therefore, we rely primarily on **average scores across multiple benchmarks** for a more stable evaluation of the model's overall capability.
>
> As shown in Tab 4, incorporating pixel tokens increases the **average benchmark score by 1.7 points** compared to using semantic tokens alone. The improvements are particularly evident on **POPE (+3.6)**, **TextVQA (+3.0)**, and **OCRBench (+4.6)**.
>
> To further validate this observation, we conducted additional experiments on **LLaVA-OneVision**, which is trained with a much larger dataset. As shown below, adding pixel tokens consistently improves performance across benchmarks:
>
> |        |Res.|MMB |MME |SEED|VQAv2|MMVet|AI2D|MMMU|POPE|OCRB|TVQA|AVG.
> |:------:|:--:|:--:|:--:|:--:|:---:|:---:|:--:|:--:|:--:|:--:|:--:|:--:|
> sem.     |384 |74.0|74.1|68.2|88.1 |39.7 |75.5|38.6|84.8|65.9|60.8|67.0
> sem.+pix.|384 |75.8|77.0|69.4|88.8 |41.3 |78.4|39.8|87.2|69.5|64.4|69.2
>
> It is worth noting the *improvements from pixel tokens are most pronounced in OCR and hallucination-related benchmarks*, where fine-grained visual details are critical, while gains are smaller on knowledge- and reasoning-oriented tasks.
>
> ### W3: Different LLM Scales
>
> We have added results for DualToken using two additional LLM backbones of different scales (**Qwen2.5-1.5B** and **Qwen2.5-7B**) to further demonstrate the method’s scalability and generality:
>
> |Model    |Res.|Params|  POPE  |   MMB  |  SEED  |  MMMU  |  MMVet |MathVista|   MME  |
> |:--------|:--:|:----:|:------:|:------:|:------:|:------:|:------:|:-------:|:------:|
> |DualToken|256 |1.5B  |  82.9  |  67.5  |  66.0  |  35.0  |  28.2  |  39.0   |  1465  |
> |DualToken|256 |3B    |  86.0  |  70.9  |  70.2  |  38.6  |  32.5  |  46.5   |  1489  |
> |DualToken|256 |7B    |**88.6**|**74.9**|**71.8**|**45.8**|**40.5**|**55.8** |**1502**|
>
> |Model          |Params|GenEval |  Wise  |
> |:--------------|:----:|:------:|:------:|
> |DualToken (256)| 1.5B |  0.69  |  0.32  |
> |DualToken (256)|  3B  |  0.72  |  0.35  |
> |DualToken (256)|  7B  |**0.75**|**0.39**|
>
> Evidently, **DualToken exhibits strong and competitive performance across various LLM scales**, with clear and steady improvement as model size increases, underscoring its scalability. (please refer to Tab.5 and 10 for other compared methods)
>
> ### W4: Ablation on Layer Boundary
>
> Thank you for the insightful suggestion! We have added a detailed ablation study and discussion on this topic in **Appendix B**, where we analyze how the layer boundary is determined and its impact on performance. (The PDF will soon be updated)

---

> ### Author Response · Authors · 2025-12-04
>
> ### Supplementary Reply to W4
>
> In *Figure 7*, we provide comprehensive visualizations of **inter-layer cosine similarity** across different backbones, revealing a **consistent hierarchical pattern** across architectures, model scales, and resolutions. A cohesive high-similarity region corresponding to the shallow layers can be clearly observed.
>
> Taking *SigLIP2-SO/16-256* as an example, the boundary between low-level and high-level layers can be clearly identified between the **9th** and **10th** layers. Quantitative results further support this observation:
>
> |      Backbone     | Layer recon./total | Zero-shot |   rFID   |
> | :---------------: | :----------------: | :-------: | :------: |
> | SigLIP2-SO/16-256 |        5/27        |    81.0   |   0.70   |
> |                   |        6/27        |    81.2   | **0.66** |
> |                   |        7/27        |    81.2   |   0.69   |
> |                   |        8/27        |  **81.3** |   0.67   |
> |                   |        9/27        |    81.0   |   0.72   |
> |                   |        10/27       |    79.6   |   0.98   |
> |                   |        12/27       |    79.4   |   1.16   |
> |                   |        14/27       |    78.4   |   1.08   |
> |                   |        21/27       |    73.7   |   3.02   |
>
> As shown, when the reconstruction layer is selected within the shallow region (**below the boundary**), performance remains stable with minimal fluctuations. However, once the reconstruction layer extends **beyond the boundary**, both reconstruction and semantic performance begin to degrade noticeably.

---

### Official Review · Reviewer_PrxC · 2025-11-01

**Soundness:** 2
**Presentation:** 2
**Contribution:** 2
**Rating:** 2
**Confidence:** 5

**Summary:**

This paper proposes DualToken, a unified tokenization pipeline that decomposes visual inputs into both content tokens and discrepancy tokens, aiming to improve visual-language and vision-only tasks under the same framework. The method is evaluated on several benchmarks to show improved representation consistency and task performance.

**Strengths:**

1. The motivation to unify visual tokenization across tasks is reasonable.

2. The writing is generally clear and the architecture is presented in an organized manner.

3. Experiments cover multiple downstream tasks.

**Weaknesses:**

1. The core idea of decomposing visual features into semantic and pixel parts has been explored extensively in prior tokenizer and unified representation works (e.g., TokenFlow, UniTok, UniToken, Bagel, Show-o2, etc). The paper does not clearly articulate what is conceptually new here.

2. The experimental comparisons omit recent strong tokenizer-based or unification SOTA models. Without these, the claimed performance improvement is not convincing.

3. The paper frames itself as “unifying tokenizers,” but the design appears largely heuristic, and no strong evidence is provided that this formulation generalizes broadly or scales to high-performing unification models.

4. There is limited explanation of how or why the proposed discrepancy tokens help. The model may simply benefit from an increased token budget rather than a meaningful representational factorization.

5. Some baselines are outdated or under-optimized. The improvement margins are small, and significance is unclear.

6. The conclusion reads brief and generic. It does not help reinforce the contributions or position the work in the broader literature.

**Questions:**

1. Why were stronger modern tokenizers not included in comparisons? Without these, the results are incomplete.

2. Can the authors provide concrete justification that the content/discrepancy decomposition corresponds to meaningful semantic factors, rather than just acting as an ad-hoc feature split?

3. What is the computational overhead introduced by the two-branch tokenization versus standard single-stream tokenizers?

4. How does the method perform when scaling to higher-resolution or longer input sequences?

5. Could the authors release qualitative examples to demonstrate that discrepancy tokens capture non-trivial representation variation?

---

> ### Author Response · Authors · 2025-11-21
> **Reply Window 1: W1-W2**
>
> Thank you for your thoughtful review. We appreciate your recognition of our motivation and writing clarity.
>
> ---
>
> ### W1: What is Conceptually New?
>
> 1. **TokenFlow, UniToken, Bagel, Show-o2, Janus, etc**
>
>    These methods rely on **two heterogeneous encoders** (VAE + CLIP/SigLIP). We have discussed the limitations of this family of approaches in our *Related Work*:
>
>    > *“A straightforward way to bypass such a conflict is to employ two heterogeneous vision encoders [1,2,3]: a semantic tokenizer (e.g., CLIP) for understanding and a reconstruction-based tokenizer (e.g., VQ-VAE) for generation. Yet this design inevitably adds extra modules and structural complexity, making understanding and generation two loosely coupled systems with distinct pathways rather than a truly unified model.”*
>
>    While *TokenFlow* enforces a shared mapping to bind semantics and pixels, such a forced coupling limits the representational capacity (Sec.2, L151–155), reflected in its inferior results (rFID 0.63 vs. ours 0.24). We have also added citations to *UniToken* and *Show-o2* accordingly.
>
> 2. **UniTok and QLIP**
>
>    UniTok and QLIP are the latest works that use a *single tokenizer* to handle both reconstruction and semantics, but share the same representation layer for both tasks.
>    Our key finding, however, reveals that using *shallow layers* for pixel yields better reconstruction performance while causing less damage to *deep-layer* semantic performance.
>    Hence, while related in topic, we are **conceptually unique: we are the first and only work to explicitly propose shallow-for-reconstruction and deep-for-semantics**, which forms the core motivation and novelty of DualToken. The superiority of our method is also validated in Tab.1.
>    Moreover, as discussed in Appendix C, **DualToken and UniTok are in fact complementary**: *UniTok* focuses on the VQ method itself (MCQ), whereas our emphasis lies on structural design. Introducing our design to UniTok brings consistent improvements in both reconstruction fidelity and zero-shot classification. Finally, we would like to kindly clarify that our work and UniTok are in fact concurrent.
>
> ### W2: Recent Tokenizer or Unification Models
>
> To the best of our knowledge, **UniTok** represents the strongest tokenizer-based approach prior to our submission. We provided comprehensive comparisons with UniTok in **Tab 1 and 5**, and also included **QLIP**, another recent tokenizer, in *Tab.1*. To further strengthen completeness, we have now added comparisons with **Muse-VL** and another concurrent tokenizer work, **TokLIP**, in the revised *Tab.1*.
>
> Method|Res.|zero-shot|rFID
> |:-:|:-:|:-:|:-:|
> QLIP|392|79.1|1.46
> Muse-VL|256|-|2.26
> TokLIP|256|76.4|2.19
> TokLIP|384|80.0|2.19
> DualToken|256|82.3|0.52
>
> In addition, we have extended our comparisons with the aforementioned works across understanding and generation benchmarks in *Tab.5(a)* and *Tab.10*, where **DualToken consistently achieves leading performance** across diverse evaluations.
>
> ||Res.|Params|POPE|MMB |SEED|MMMU|MMVet|MathVista|MME
> |:-:|:-:|:-:|:-:|:-:|:-:|:-:|:-:|:-:|:-:|
> TokenFlow|256|13B|85.0|60.3|62.6|34.4|27.7|-|1365
> TokenFlow|384|13B|86.8|68.9|68.7|38.7|40.7|-|1546
> Muse-VL|256|7B|-|72.1|69.1|39.7|-|51.3|1480
> TokLIP|384|7B|84.9|76.9|70.4|43.1|29.8|- |1488
> UniTok|256|7B|83.2|-|-|-|33.9|-|1448
> UniToken|384|7B|-|71.1|69.9|32.8|-|38.5|-
> show-o2|432|1.5B|-|67.4|65.6|37.1|-|-|1451
> show-o2|432|7B|-|79.3|69.8|48.9|-|-|1621
> DualToken|256|3B|86.0|70.9|70.2|38.6|32.5|46.5|1489
> DualToken|384|3B|88.1|76.2|72.2|40.3|40.2|49.2|1588
> DualToken|256|7B|88.6|74.9|71.8|45.8|40.5|55.8|1502
> DualToken|384|7B|89.4|80.0|75.5|47.4|44.3|57.6|1625
>
> ||Params|GenEval|Wise
> |:-:|:-:|:-:|-:|
> TokenFlow (256)|13B|0.63|-
> Muse-VL (256)|7B|0.57|-
> UniTok (256)|7B|-|-
> UniToken (512)|7B|0.63|-
> show-o2 (432)|1.5B|0.73|0.35
> show-o2 (432)|7B|0.76|0.39
> DualToken (256)|3B|0.72|0.35
> DualToken (256)|7B|0.75|0.39
>
> As for other recent unification models such as **Bagel**, Mogao, Qwen-Image (and Show-o2), these fall under the scope of **AR+Diffusion** architecture (with external diffusion decoder or flow head). In contrast, our work focuses on Chameleon-style pure autoregressive (AR) architectures, which provide the most comparable and meaningful setting for evaluating *visual tokenization*. In this scope, **DualToken achieves SOTA performance** among existing unified AR models, including **UniTok** and **MUSE-VL**.
>
> Moreover, *Bagel, Mogao, and Qwen-Image* are *industrial-scale models* that leverage massive data and incorporate extensive optimizations across multiple components (e.g., MoT), which are beyond the scope of this paper. For completeness, we are open to including these models in our comparison tables with appropriate annotations.

---

> ### Author Response · Authors · 2025-11-21
> **Reply Window 2: W3-W4**
>
> ### W3: Generalization and Performance
>
> **1.** We would like to respectfully clarify that a *unified tokenizer* refers to a single tokenizer handling both reconstruction and semantic objectives, without assuming any specific implementation strategy. (e.g., whether or not heuristics are used)
>
> **2. Generalization**
>
> We have validated the generalizability of our method across multiple backbones (*SigLIP-L/16-256*, *SigLIP-SO/14-384*, and *SigLIP2-SO/16-256*) in the main text and further extended it to hybrid CNN-Transformer architectures (**ViTamin**) in **Appendix B**.
>
> To further strengthen our claims, we now include a **comprehensive validation** of the method’s generalizability, covering five mainstream backbones—**OpenAI's CLIP**, **Apple's DFN**, **BAAI's EVA**, **Google's SigLIP**, and **SigLIP2**—under consistent settings (RVQ=8, codebook size=16,384, 2M training samples). Our findings are summarized as follows.
>
> **(1) Consistent optimal layer range across architectures.**
>
> For all tested backbones, selecting the **first quarter of layers** for reconstruction consistently yields the best reconstruction quality and semantic performance:
>
> 1) Ablation on model backbone
> |Backbone|Layer recon./total|Zero-shot|rFID|
> |:--:|:--:|:--:|:--:|
> |DFN-L/14-224|6/24|**79.2**|**0.84**|
> ||12/24|77.1|1.35|
> ||18/24|72.2|3.25|
> |EVA-02-L/14-224|6/24|**77.8**|**0.80**|
> ||12/24|75.2|1.16|
> ||18/24|69.9|3.22|
> |CLIP-L/14-224|6/24|**73.2**|**0.87**|
> ||12/24|70.8|1.80|
> ||18/24|65.5|3.58|
> |SigLIP-L/16-256|6/24|**78.8**|**0.78**|
> ||12/24|76.3|1.27|
> ||18/24|72.9|2.61|
> |SigLIP2-L/16-256|6/24|**80.4**|**0.72**|
> ||12/24|77.6|1.09|
> ||18/24|72.9|2.93|
> |ViTamin-XL-384|6/24|**80.0**|**0.39**|
> ||12/24|78.6|0.88|
> ||18/24|73.8|2.25|
>
> 2) Ablation on model size & total layer
> |Backbone|Layer recon./total|Zero-shot|rFID|
> |:--:|:--:|:--:|:--:|
> |DFN-B/16-224|3/12|**74.1**|**0.98**|
> ||6/12|71.4|1.58|
> ||9/12|66.1|3.21|
> |DFN-L/14-224|6/24|**79.2**|**0.84**|
> ||12/24|77.1|1.35|
> ||18/24|72.2|3.25|
> |DFN-H/14-224|8/32|**81.0**|**0.73**|
> ||16/32|79.5|1.28|
> ||24/32|74.3|2.78|
> |SigLIP2-L/16-256|6/24|**80.4**|**0.72**|
> ||12/24|77.6|1.09|
> ||18/24|72.9|2.93|
> |SigLIP2-SO/16-256|7/27|**81.2**|**0.69**|
> ||14/27|78.4|1.08|
> ||21/27|73.7|3.02|
>
> **(2) Robustness to specific layer choice.**
>
> Our method remains stable as long as the reconstruction layers lie roughly within the first quarter of the network. Minor shifts (±1–2 layers) cause negligible changes, confirming its robustness and broad applicability:
>
> |Backbone|Layer recon./total|Zero-shot|rFID|
> |:--:|:--:|:--:|:--:|
> |DFN-B/16-224|3/12|**74.1**|0.98|
> ||4/12|73.9|**0.97**|
> |DFN-L/14-224|5/24|79.0|**0.82**|
> ||6/24|**79.2**|0.84|
> ||7/24|78.9|0.84|
> |DFN-H/14-224|6/32|**81.1**|0.73|
> ||8/32|81.0|0.73|
> ||10/32|80.7|**0.72**|
> |SigLIP2-L/16-256|5/24|80.4|0.74|
> ||6/24|**80.4**|**0.72**|
> ||7/24|80.2|0.75|
> |SigLIP2-SO/16-256|5/27|81.0|0.70|
> ||6/27|81.2|**0.66**|
> ||7/27|81.2|0.69|
> ||8/27|**81.3**|0.67|
> ||9/27|81.0|0.72|
>
> These findings indicate that for a new ViT architecture, one can confidently select the first quarter of layers for reconstruction to achieve optimal results.
>
> Moreover, in **Appendix.B**, we also provide mroe visualizations of inter-layer cosine similarity across backbones, revealing a **consistent hierarchical pattern** divisible into shallow and deep layers (also supported by prior study [a]), further supporting the universality of our method.
>
> **2. Performance**
>
> We report the performance of DualToken from 3B to 7B in Tab 5 and 10 (see also *Reply Window 1*), where it achieves **state-of-the-art performance among all AR-based unified models**. Therefore, we respectfully disagree with the reviewer's comment regarding the absence of "high-performing unification models."
>
> ### W4: Token Budget v.s. Factorization
>
> Thank you for this question.
>
> 1. In **Tab 3(a)**, the *Recon.(26) + Sem.(26)* setting attaches two codebooks at the 26th layer (each with RVQ*8, codebook size 16384). One is quantized and fed to the reconstruction branch, and the other to the semantic branch. Thus, the overall token budget is identical to that of DualToken. However, its performance is significantly worse, demonstrating that the key improvement does NOT arises from increased token budget. We have added clarification of this setting in the revised version.
>
> 2. To further isolate this effect, we conducted an additional experiment where we removed VQ and trained the model directly in the continuous space (i.e., with an effectively infinite token budget). The results show that learning reconstruction in the shallow layer and semantics in the deep layer still yields substantial improvement. This confirms that the conflict between reconstruction and semantics inherently exists in the continuous representation space, and that **layer-wise factorization** provides a meaningful way to resolve this conflict.
>
> |Setting|Zero-shot|rFID|
> |:--:|:--:|:--:|
> |Recon.(26) + Sem.(26)|79.2|1.27|
> |Recon.(6) + Sem.(26)|82.9|0.19|

---

> ### Author Response · Authors · 2025-11-21
> **Reply Window 3: W5-W6, Q1-Q5**
>
> ### W5: Baselines and Improvement Margins
>
> 1. **Baselines**
>
>     We respectfully disagree.
>
>    * "Outdated": **All compared methods are recent works from within the past year**, including **UniTok**, **TokenFlow**, and **TokLIP**, which represent the latest and most relevant advances in this direction. As UniTok, our model, like theirs, is built upon VILA-U. Therefore, dedicating more discussion to comparisons against VILA-U provides the most direct and fair demonstration of our contribution.
>
>    * "Under-optimized": We would like to clarify that our baseline design is carefully implemented and fully documented in **Appendix F: Implementation Details – Dual Encoder Baseline**. The *sole purpose* of this experiment is to isolate the effect of the **source of dual visual tokens**—i.e., comparing dual visual vocabularies obtained *within a unified tokenizer* v.s. *from two heterogeneous tokenizers*.
>
>       To ensure strict fairness and control, we standardized all other factors:
>       * Identical image resolution, token length (16*16), RVQ depth (D=4), embedding dimension, model architecture, and training data.
>       * The **only difference** lies in how visual vocabularies are obtained:
>          * Ours: both pixel-level and semantic-level codebooks are derived from different layers of the **same vision encoder**.
>          * Baseline: pixel and semantic codebooks come from two separate encoders (VQGAN + SigLIP).
>       * Both encoders in the dual-encoder baseline are well trained on the same data as DualToken and perform competitively on their respective tasks:
>
>       ||Zero-shot|rFID|
>       |:--:|:--:|:--:|
>       |SigLIP-VQ|80.6|–|
>       |VQGAN|–|1.25|
>
>       These controls ensure the comparison isolates the tokenization design itself, independent of data quality, training setup, or model architecture.
>
> 2. **Improvement Margins**
>
>    Regarding the comment that the *improvement margins are small*, we kindly ask the reviewer to clarify which specific results this refers to. Across the majority of benchmarks, DualToken consistently achieves statistically meaningful gains over compared methods.
>
> ### W6
>
> Thank you for the feedback! The conclusion section has been revised, and the updated PDF manuscript will be uploaded soon.
>
> ### Q1
>
> See W2.
>
> ### Q2
>
> See W4 and Q5.
>
> ### Q3: Computational Overhead
>
> As has been discussed in Appendix.D, introducing two codebooks **DOES NOT** significantly increase the computational overhead:
>
> **1. Parameter Count**
>
> The ONLY additional parameters arise from 3 components:
> - The MLP projector's hidden_dim change from (1024->2048->2048) to (2048->2048->2048), which adds **2.1M** parameters.
> - An additional visual head: **258M** parameters.
> - An additional VQEmbedding layer: **16M** parameters.
>
> Together, these account for only **0.089** of the total parameters compared to the LLM backbone (3B). When scaling to larger backbones (e.g., 7B), the relative impact becomes even more negligible.
>
> **2. Memory Usage and Inference Latency.**
>
> Since our dual tokens are concatenated along **feature dimension** rather than **sequence dimension**, and the input dimension to the LLM remains unchanged, **no new pathway is introduced to the LLM**, and the computational cost of the LLM backbone remains **strictly the same**. The only increase stems from the components listed above.
>
> ||Training Memory Usage|Inference Time|Single Forward GFLOPs
> |:--:|:--:|:--:|:--:|
> single token|73.8G|11.42s|328.98
> dual token|78.2G|12.97s|337.20
>
> Memory usage is measured under the same local batch size and device. FLOPs and inference time are averaged on T2I task (256px) over the MJHQ-30K dataset.
>
> ### Q4
>
> Thanks for the question. Increasing image resolution leads to consistent improvements in understanding performance, as shown in Tab.5(a)
>
> For generation tasks, scaling up the resolution also yields notable gains on aesthetics-oriented benchmarks like MJHQ, aligning with the observations reported in VILA-U.
>
> Models (7B)|Res.|MJHQ
> |:--:|:--:|:--:|
> Muse-VL|256|7.73
> Muse-VL|384|-
> UniTok|256|7.46
> UniTok|384|-
> VILA-U|256|12.81
> VILA-U|384|7.69
> DualToken|256|7.25
> DualToken|384|5.96
>
> ### Q5: On Qualitative Examples of Discrepancy Tokens
>
> We appreciate the reviewer's insightful suggestion. We have added qualitative examples in the **Appendix**, including:
>
> 1. **Clustering results** obtained using the discrete representations of *pixel* and *semantic* tokens;
> 2. Visualization of feature maps
>
> [a] Chen H, et al. Rethinking Visual Layer Selection in Multimodal LLMs. 2025.

---

> > ### Comment · Reviewer_PrxC · 2025-11-26
> > **Thanks for your detailed response.**
> >
> > I thank the authors for their efforts.
> >
> > For weakness 1, I am concerned about the novelty of the technical contributions. The author provides detailed explanations about the difference with prior studies (UniTok, etc) with shallow-deep combination for both pixel and semantic representations. My concerns are mostly resolved.
> >
> > For weakness 2, in the pure AR category, the proposed DualToken outperforms other counterparts. I understand that there is still a gap compared to Bagel, Mogao, and Qwen-Image. The author can try to display it in gray in the table.  Overall, most of my concerns are addressed with added comparisons.
> >
> > For weakness 3-4, the newly added experiments demonstrate strong and consistent performance gains with layer selection. Besides, a further explanation of layer-wise factorization addresses my related concerns.
> >
> > Other issues include improved margin and computing costs. In particular, this dual-codebook strategy does not seem to lead to excessive consumption of training and testing. This point is highly favored.
> >
> > Thank you once again to the author for the further clarification.  All of my major concerns have been resolved, I am increasing my score to an acceptance.

---

> ### Author Response · Authors · 2025-11-28
> **Thanks for Your Review and Raising the Score**
>
> Thank you very much for your encouraging feedback and for raising the score from 2 to 6. We're glad our responses have addressed your concerns. We will incorporate the additional results into the revised version. Once again, we sincerely appreciate your time and insightful review!

---

### Official Review · Reviewer_AqP7 · 2025-11-01

**Soundness:** 3
**Presentation:** 3
**Contribution:** 3
**Rating:** 6
**Confidence:** 4

**Summary:**

The paper proposes DualToken, a unified visual tokenizer that decouples the tokenization process by applying quantization to shallow and deep layers of a single SigLIP backbone for reconstruction and semantic learning, respectively. The resulting tokenizer demonstrates strong reconstruction performance and achieves consistent improvements over VILA-U across both visual understanding and generation tasks.

**Strengths:**

- The proposed method demonstrates solid results across both understanding and generation tasks.
- The paper is well written and easy to follow.

**Weaknesses:**

1. Insufficient Experiments: The paper would benefit from more comprehensive ablations on model design, such as analyzing how the choice of reconstruction layer affects the final performance.
2. Limited Baseline Comparison: The single-encoder vs. dual-encoder experiment lacks comparison with stronger dual-branch architectures, which undermines the validity of the method’s claimed effectiveness.

**Questions:**

1. Table 1 would benefit from a clearer organization, for example by grouping models according to factors such as the number of model parameters, downsampling ratio or number of tokens, codebook size.
2. The experiment of comparing single-encoder versus dual-encoder designs would be more convincing if compared against recent dual-branch architectures such as TokenFlow[1] or MuseVL[2], which also aim to reconcile semantic and reconstruction requirements rather than merely stacking two unrelated encoders.

[1] Qu, Liao, et al. "Tokenflow: Unified image tokenizer for multimodal understanding and generation." Proceedings of the Computer Vision and Pattern Recognition Conference. 2025.

[2] Xie, Rongchang, et al. "Muse-vl: Modeling unified vlm through semantic discrete encoding." Proceedings of the IEEE/CVF International Conference on Computer Vision. 2025.

---

> ### Author Response · Authors · 2025-11-21
>
> We sincerely thank the reviewer for the constructive and positive feedback. We are glad that you found our work *demonstrates solid results*, and is *well written* and *easy to follow*.
>
> ---
>
> ### W1: More Ablations
>
> Thank you for the valuable comment. We have added detailed experiments analyzing how the choice of reconstruction layer impacts both the tokenizer and the downstream multimodal performance. The corresponding results and visualizations are included in **Appendix B**.
>
> 1. Effect on tokenizer performance
>
> We evaluate six representative backbones—**OpenAI's CLIP**, **Apple's DFN**, **BAAI's EVA**, **Google's SigLIP, SigLIP2** and **ViTamin**—under consistent settings (RVQ=8, codebook size=16,384, 2M training samples).
>
> Across all tested backbones, selecting the **first quarter of layers** for reconstruction consistently yields the best reconstruction quality and semantic performance. As deeper layers are used, the conflict becomes increasingly evident, degrading both capabilities:
>
> |Backbone         |Layer recon./total|Zero-shot|rFID|
> |:---------------:|:----------------:|:-------:|:--:|
> |DFN-L/14-224     |6/24              |**79.2** |**0.84**|
> |                 |12/24             |77.1     |1.35|
> |                 |18/24             |72.2     |3.25|
> |EVA-02-L/14-224  |6/24              |**77.8** |**0.80**|
> |                 |12/24             |75.2     |1.16|
> |                 |18/24             |69.9     |3.22|
> |CLIP-L/14-224    |6/24              |**73.2** |**0.87**|
> |                 |12/24             |70.8     |1.80|
> |                 |18/24             |65.5     |3.58|
> |SigLIP-L/16-256  |6/24              |**78.8** |**0.78**|
> |                 |12/24             |76.3     |1.27|
> |                 |18/24             |72.9     |2.61|
> |SigLIP2-L/16-256 |6/24              |**80.4** |**0.72**|
> |                 |12/24             |77.6     |1.09|
> |                 |18/24             |72.9     |2.93|
> |ViTamin-XL-384   |6/24              |**80.0** |**0.39**|
> |                 |12/24             |78.6     |0.88|
> |                 |18/24             |73.8     |2.25|
>
> Moreover, our method also shows remarkable robustness to specific layer choice, as long as the reconstruction layers lie roughly within the first quarter of the network. Minor shifts (±1–2 layers) cause negligible changes, confirming its robustness and broad applicability:
>
> ||Layer recon./total|Zero-shot|rFID|
> |:----:|:----:|:---:|:---:|
> ||5/24|80.4|0.74|
> |SigLIP2-L/16-256|6/24|**80.4** |**0.72**|
> ||7/24|80.2|0.75|
>
> 2. Effect on downstream understanding and generation
>
> We further trained 3B-scale unified models on half of the data to examine how reconstruction-layer depth affects downstream performance.
>
> **(1) Understanding**
>
> |Layer recon./total|Zero-shot|POPE |MMB  |SEED |MMMU |TVQA
> |:-----:|:-------:|:---:|:---:|:---:|:---:|:---|
> |6/24  |**80.4**|**83.6**|**69.8**|**69.2**|**35.5**|**55.8**
> |12/24|77.6     |80.1 |68.4 |63.6 |33.6 |50.2
> |18/24|72.9     |74.5 |64.1 |58.8 |28.8 |40.8
>
> Downstream VQA understanding correlates strongly with zero-shot accuracy.
>
> **(2) Generation**
>
> |Layer recon./total|Zero-shot|rFID    |GenEval|MJHQ |
> |:-------:|:-------:|:------:|:-----:|:---:|
> |6/24     |**80.4**|**0.72**|**0.67**|**10.21**
> |12/24   |77.6     |1.09    |0.65   |12.75
> |18/24   |72.9     |2.93    |0.58   |17.87
>
> As the reconstruction layer deepens, the reconstruction quality deteriorates, leading to blurred and distorted generated images. While this degradation is less pronounced in semantic metrics such as GenEval, it is much more evident in aesthetic-related benchmarks (e.g., MJHQ).
>
> ### W2: More Comparisons against Recent Dual-branch Architectures
>
> Thank you for the valuable suggestion. We have added comparisons against recent *dual-branch architectures* including **TokenFlow**, **MuseVL** and **UniToken**[a] in **Tab 5 and 10**.
>
> ||Res.|Params|POPE|MMB |SEED|MMMU|MMVet|MathVista|MME
> |:-:|:-:|:-:|:-:|:-:|:-:|:-:|:-:|:-:|:-:|
> TokenFlow|256|13B|85.0|60.3|62.6|34.4|27.7|-|1365
> Muse-VL|256|7B|-|72.1|69.1|39.7|-|51.3|1480
> UniToken|384|7B|-|71.1|69.9|32.8|-|38.5|-
> DualToken|256|3B|86.0|70.9|70.2|38.6|32.5|46.5|1489
> DualToken*|256|7B|88.6|74.9|71.8|45.8|40.5|55.8|1502
>
> ||Params|GenEval|Wise
> |:-:|:-:|:-:|-:|
> TokenFlow (256)|13B|0.63|-
> Muse-VL (256)|7B|0.57|-
> UniToken (512)|7B|0.63|-
> DualToken (256)|3B|0.72|0.35
> DualToken* (256)|7B|0.75|0.39
>
> *: We scaled DualToken to 7B in response to reviewer kK8G.
>
> These results demonstrate that **DualToken consistently outperforms dual-branch architectures** on both understanding and generation tasks, highlighting the effectiveness of our single-encoder, dual-codebook design.
>
> ### Q1: Organization of Tab.1
> Thanks for the suggestion! We will update the table in the PDF.
>
> ### Q2
> See W2.
>
> [a] Jiao, Yang, et al. "Unitoken: Harmonizing multimodal understanding and generation through unified visual encoding." 2025.

---

### Official Review · Reviewer_GZrq · 2025-11-01

**Soundness:** 3
**Presentation:** 3
**Contribution:** 3
**Rating:** 6
**Confidence:** 4

**Summary:**

This paper proposes DualToken, a novel approach aimed at unifying visual understanding and generation within autoregressive large language models (LLMs). The core idea is to introduce dual visual vocabularies—one for low-level visual details (used in generation) and another for high-level semantics (used in understanding). By decoupling these two objectives, the authors resolve the conflict between reconstruction and semantic tasks, leading to improved performance in both areas. The approach also shows strong effectiveness in multimodal understanding benchmarks, making it a promising solution for unified vision-language models. The method is efficient, with minimal additional computational overhead, and the results highlight the advantages of unifying visual understanding and generation in a single framework.

**Strengths:**

The writing in this paper is clear and well-structured, effectively explaining the novel concept of dual visual vocabularies and its benefits in unifying visual understanding and generation. The experimental design is robust, with thorough evaluations across multiple benchmarks, demonstrating the superiority of DualToken over existing methods in both reconstruction and semantic tasks. The authors also provide comprehensive comparisons and detailed ablation studies, which strengthen the validity of their claims and offer valuable insights into the practical impact of their approach.

**Weaknesses:**

1.Heuristic layer selection limits generalization and reusability. The method applies reconstruction supervision to early layers and semantic supervision to a fixed deep layer, yet this design choice lacks thorough ablation studies and relies heavily on heuristics, raising concerns about its generalizability across architectures and tasks.

2.Overly simplified semantic supervision may undermine representation quality. The use of L2 regression to pretrained SigLIP’s final-layer features, without contrastive objectives or cosine similarity constraints, raises questions about whether this feature-level alignment is sufficient to preserve rich semantic capacity.

**Questions:**

Please refer to the above weakness.

---

> ### Author Response · Authors · 2025-11-21
>
> We sincerely thank the reviewer for the thoughtful and positive feedback. We are delighted that you found our paper:
> * presents a novel concept with clear and well-structured writing,
> * demonstrates robust experimental design, and
> * provides comprehensive comparisons and detailed ablations that offer valuable insights into the practical impact of our approach.
>
> ---
>
> ### W1: Layer Selection and Generalizability
>
> We have validated the generalization and reusability of our method across multiple backbones (**SigLIP-L/16-256**, **SigLIP-SO/14-384**, and **SigLIP2-SO/16-256**) in the main text and further extended it to hybrid CNN–Transformer architectures (**ViTamin**) in **Appendix B**.
>
> To make it more concrete, we include a **comprehensive validation** of the method's generalizability, covering six mainstream backbones—**OpenAI's CLIP**, **Apple's DFN**, **BAAI's EVA**, **Google's SigLIP, SigLIP2** and **ViTamin**—under consistent settings (RVQ=8, codebook size=16384). Our findings are summarized as follows.
>
> 1. **Consistent optimal layer across architectures.**
>    For all tested backbones, selecting the **first quarter of layers** for reconstruction consistently yields the best reconstruction quality and semantic performance:
>
> (1) ablation on model backbone
> |Backbone|Layer recon./total|Zero-shot|rFID|
> |:--:|:--:|:--:|:--:|
> |DFN-L/14-224|6/24|**79.2**|**0.84**|
> ||12/24|77.1|1.35|
> ||18/24|72.2|3.25|
> |EVA-02-L/14-224|6/24|**77.8**|**0.80**|
> ||12/24|75.2|1.16|
> ||18/24|69.9|3.22|
> |CLIP-L/14-224|6/24|**73.2**|**0.87**|
> ||12/24|70.8|1.80|
> ||18/24|65.5|3.58|
> |SigLIP-L/16-256|6/24|**78.8**|**0.78**|
> ||12/24|76.3|1.27|
> ||18/24|72.9|2.61|
> |SigLIP2-L/16-256|6/24|**80.4**|**0.72**|
> ||12/24|77.6|1.09|
> ||18/24|72.9|2.93|
> |ViTamin-XL-384|6/24|**80.0**|**0.39**|
> ||12/24|78.6|0.88|
> ||18/24|73.8|2.25|
>
> (2) ablation on model size & total layer
> |Backbone|Layer recon./total|Zero-shot|rFID|
> |:--:|:--:|:--:|:--:|
> |DFN-B/16-224|3/12|**74.1**|**0.98**|
> ||6/12|71.4|1.58|
> ||9/12|66.1|3.21|
> |DFN-L/14-224|6/24|**79.2**|**0.84**|
> ||12/24|77.1|1.35|
> ||18/24|72.2|3.25|
> |DFN-H/14-224|8/32|**81.0**|**0.73**|
> ||16/32|79.5|1.28|
> ||24/32|74.3|2.78|
> |SigLIP2-L/16-256|6/24|**80.4**|**0.72**|
> ||12/24|77.6|1.09|
> ||18/24|72.9|2.93|
> |SigLIP2-SO/16-256|7/27|**81.2**|**0.69**|
> ||14/27|78.4|1.08|
> ||21/27|73.7|3.02|
>
> 2. **Robustness to specific layer choice.**
>    Our method remains stable as long as the reconstruction layers lie roughly within the first quarter of the network. Minor shifts (±1–2 layers) cause negligible changes, confirming its robustness and broad applicability:
>
> |Backbone|Layer recon./total|Zero-shot|rFID|
> |:--:|:--:|:--:|:--:|
> |DFN-B/16-224|3/12|**74.1**|0.98|
> ||4/12|73.9|**0.97**|
> |DFN-L/14-224|5/24|79.0|**0.82**|
> ||6/24|**79.2**|0.84|
> ||7/24|78.9|0.84|
> |DFN-H/14-224|6/32|**81.1**|0.73|
> ||8/32|81.0|0.73|
> ||10/32|80.7|**0.72**|
> |SigLIP2-L/16-256|5/24|80.4|0.74|
> ||6/24|**80.4**|**0.72**|
> ||7/24|80.2|0.75|
> |SigLIP2-SO/16-256|5/27|81.0|0.70|
> ||6/27|81.2|**0.66**|
> ||7/27|81.2|0.69|
> ||8/27|**81.3**|0.67|
> ||9/27|81.0|0.72|
>
> The findings suggests that, for a new vision backbone, one can confidently select the first quarter of layers for reconstruction to achieve ideal results, **demonstrating strong generalization and reusability across architectures.**
>
> We have included this detailed experiment in **Appendix.B**, where we also provide additional visualizations of inter-layer cosine similarity across 6 backbones (Fig.7). It shows that all backbones share the **same hierarchical pattern** of shallow and deep layers (also supported by prior study [a]), further reinforcing the universality of our method.
>
> ### W2: Semantic Supervision
>
> Sorry for the confusion. In fact, we use a combination of L2 and cosine similarity loss (see attached code). We have clarified the description in the paper to avoid any potential confusion.
>
> We would also like to clarify the following:
>
> 1. **Strong downstream performance.** The most reliable indicator of representation quality lies in downstream visual understanding performance. As shown in Tables 4 and 5(a), DualToken achieves superior results across VQA benchmarks, clearly demonstrating the high quality of its representations.
> 2. **Adding cosine similarity loss yields only MINOR differences.** Cosine similarity and L2 alignment are inherently similar. Both aim to bring two vectors closer in space (cosine similarity focuses on angle, while L2 accounts for both magnitude and angle). Therefore, it is not a crucial part of our method.
> 3. **Contrastive learning (CL) loss** performs even worse than L2+cosine similarity loss under our limited data scale. Direct feature alignment offers a simpler and more efficient approach, and only needs unlabeled image data.
>
> Below we provide our early ablation results on SigLIP-SO400M/14-384.
>
> |Loss|Zero-shot|Avg. Score on LLaVA-1.5
> |:--:|:--:|:--:|
> |L2|80.0|52.6|
> |L2+cos-sim|80.6|53.2|
> |CL|79.2|51.9|
>
> [a] Chen H, et al. Rethinking Visual Layer Selection in Multimodal LLMs. 2025.

---

### Author Response · Authors · 2025-12-02
**# Summary of Rebuttal Updates and Post-Rebuttal Consensus**

Dear Reviewers, AC, SAC, and PC,

We sincerely appreciate the time and effort you devoted to reviewing our paper. We are delighted that ***all reviewers*** recognized the value of *DualToken* and ***recommended acceptance*** during the discussion phase ***before November 27*** (prior to the OpenReview bug, the post-rebuttal score was ***6666***).

---

### **1. Reviewer Engagement & Score Updates (6626 $\rightarrow$ 6666)**

Firstly, we are deeply grateful to the reviewers for recognizing that our paper:

* proposes a method that is ***elegant and effective*** (*kK8G*);
* presents a ***well-articulated motivation with a clear and logical flow*** (*kK8G*, *PrxC*);
* is ***thoroughly validated*** through comprehensive experiments (*kK8G*), demonstrates *robust experimental design and thorough evaluations* (*GZrq*), and ***achieves solid results*** (*AqP7*);
* provides *comprehensive comparisons and detailed ablations* that *offer valuable insights* into the practical impact of our approach (*GZrq*); and
* introduces a ***novel concept*** of *dual visual vocabularies* (*GZrq*) with ***clear and well-structured writing*** (*GZrq*, *AqP7*, *PrxC*).

We have made every effort to address each concern raised by the reviewers and significantly strengthened the manuscript.
In particular, we successfully addressed the concerns of Reviewer *PrxC*, who subsequently ***RAISED the score to an ACCEPTANCE (Score 2 $\rightarrow$ 6)*** at 15:13 on Nov 25, 2025 (AOE).

---

### **2. Key improvements**

1. **Detailed Ablations on Generalizability and Model Design**

   * **Ablation on model backbone:** We exhaustively verified the generalizability of our method across six mainstream ViT architectures (*GZrq*, *PrxC*).
   * **Ablation on model size and total layers:** We demonstrated that our layer-selection strategy generalizes well to ViTs of different scales (*GZrq*, *PrxC*).
   * **Robustness of layer selection:** We confirmed that model performance remains stable under small variations in the selected layer (*GZrq*, *PrxC*).
   * **Impact of specific layer choices:** We further analyzed how the chosen reconstruction layer affects downstream understanding and generation tasks (*AqP7*).

2. **Scalability**: We further validated the scalability of DualToken at the *1.5B* and *7B* model scales (*kK8G*).

3. **Comprehensive SOTA Benchmarking**: We added comparisons with recent *dual-branch architectures* (*AqP7*) and the latest *tokenization and unification models* (*PrxC*), including *UniTok*, *Muse-VL*, *TokLIP*, and *Show-o2*, where DualToken achieves ***SOTA performance*** among autoregressive unified models, further reinforcing DualToken's leading performance.

4. **Clarified Core Novelty**: We provided clearer distinctions between our key insight and several concurrent works (*PrxC*). To the best of our knowledge, we are the ***FIRST*** study to explicitly propose the novel concept of *dual visual vocabularies*, and the *shallow-for-reconstruction, deep-for-semantics* principle for image tokenizers.

5. **Token Budget vs. Representation Factorization**: By clarifying our experimental setup—where the token budget is strictly controlled to be identical—and conducting direct experiments in continuous space, we confirm that the improvement stems from *meaningful representational factorization* rather than token budget. This successfully addresses *PrxC*'s related concern.

6. **Computational Efficiency**: As detailed in *Appendix D*, we show that the additional computational cost is minimal. Reviewer *PrxC* explicitly acknowledged this, noting:

   > *"In particular, this dual-codebook strategy does not lead to excessive consumption of training or testing resources. This point is highly favored."*

We believe these revisions have further strengthened the paper and clarified our contributions.

---

Once again, we thank all reviewers for their efforts and constructive feedback, and we are grateful to the AC for the coordination, organization, and professional evaluation.

Best regards,

*Authors of Paper #16577*

---

### Meta-Review · Area_Chair_cgWp · 2026-01-06

**Summary:**

This paper proposes DualToken, a unified visual tokenizer with dual visual vocabularies for autoregressive multimodal LLMs. The submission initially received mixed ratings (6, 6, 6, 2). The Reviewer PrxC recommended rejection with a score of 2, with primary concerns focused on the novelty of the method, potential computational overhead, generalization capabilities, and a lack of comparisons with state-of-the-art tokenizers. During the rebuttal phase, the authors provided a comprehensive response that effectively resolved all these issues. Moreover, during the rebuttal period, Reviewer PrxC acknowledged the improvements and raised the score to recommend acceptance.

After a careful review of the paper and the rebuttal communication, the AC finds that the authors have effectively addressed all raised concerns. With all positive scores, the AC aligns with all the reviewers and recommends acceptance.

**Reviewer Concerns:**

All concerns have been addressed by the rebuttal.

**Reviewer Scores:**

Reviewer PrxC confirmed that all concerns were addressed and increased the score to a positive rating.

---

### Decision · Program_Chairs · 2026-01-26

Accept (Poster)